**communications**

**biology**

# Multivariate EEG activity reflects the Bayesian integration and the integrated Galilean relative velocity of sensory motion during sensorimotor behavior

Woojae Jeong[1,2,6], Seolmin Kim[1,3,6], JeongJun Park[1,4] & Joonyeol Lee [1,3,5✉]

Humans integrate multiple sources of information for action-taking, using the reliability of each source to allocate weight to the data. This reliability-weighted information integration is a crucial property of Bayesian inference. In this study, participants were asked to perform a smooth pursuit eye movement task in which we independently manipulated the reliability of pursuit target motion and the direction-of-motion cue. Through an analysis of pursuit initiation and multivariate electroencephalography activity, we found neural and behavioral evidence of Bayesian information integration: more attraction toward the cue direction was generated when the target motion was weak and unreliable. Furthermore, using mathematical modeling, we found that the neural signature of Bayesian information integration had extra-retinal origins, although most of the multivariate electroencephalography activity patterns during pursuit were best correlated with the retinal velocity errors accumulated over time. Our results demonstrated neural implementation of Bayesian inference in human oculomotor behavior.

[1] Center for Neuroscience Imaging Research, Institute for Basic Science (IBS), Suwon 16419, Republic of Korea. [2] Department of Biomedical Engineering, University of Southern California, Los Angeles, CA 90089, USA. [3] Department of Biomedical Engineering, Sungkyunkwan University, Suwon 16419, Republic of Korea. [4] Division of Biology and Biomedical Sciences, Program in Neurosciences, Washington University in St. Louis, St. Louis, MO 63130, USA. [5] Department of Intelligent Precision Healthcare Convergence, Sungkyunkwan University, Suwon 16419, Republic of Korea. [6] These authors contributed equally: Woojae Jeong, Seolmin Kim. ✉email: joonyeol@g.skku.edu

When we interact with dynamically changing environments, our brains must rapidly process incoming motion information, make decisions, and take action. To make this process more efficient, we integrate sensory motion information with other sources to enhance the information quality. When integrated, the gain of each piece of information is adjusted according to various factors. One of these factors is the reliability of the information. Reliability-based optimal integration has been extensively studied and explained using Bayesian inference models[1]. Evidence for the dependence of accurate motor control on Bayesian inference is available in different reports regarding behavior, such as those on force estimation[2], interval timing[3,4], visual perception[5], and oculomotor processes[6].

Recent studies have shown that reliability-based optimal information integration occurs in the smooth pursuit of eye movements[7–15]. Prior knowledge accumulated by the statistical distribution of each sensory motion parameter interacts with the motion information of the tracking target and increases the bias, and it reduces the variability of the behavior. The effect of prior knowledge on behavioral response is more significant when the motion information from the tracking target is weak and unreliable (i.e., motion target with low luminance contrast). Some neural mechanisms for the reliability-based integration of information in smooth pursuit eye movements have been revealed. Studies on monkeys have shown that prior knowledge of motion stimulus in a smooth pursuit eye movement is formed and shaped by frontal cortical activity[8]. However, for human oculomotor studies, whether the optimal integration of multiple pieces of information occurs in neural activity, and guides the behavioral changes have not been fully investigated (see ref. [15] as an exception).

To determine if reliability-based information integration occurs in smooth pursuit eye movements and concurrent neural responses, we studied how direction information provided by a motion cue is combined with the pursuit target under different luminance contrasts. Consistent with the Bayesian inference prediction, the effect of cue direction on the initiation of smooth pursuit was more pronounced when the contrast of the pursuit target was low. The pursuit traces with low-contrast targets were more attracted toward the cue direction than those with high-contrast targets. The same adjustment of the gain based on the reliability of each stimulus was also observed in the multivariate electroencephalography (EEG) activity pattern. Furthermore, by modulating smooth pursuit eye movements and EEG activity using the validity of the motion direction cue and stimulus contrast, we investigated the origin of information represented in the multivariate EEG activity during the initiation of smooth pursuit behavior with mathematical modeling: most of the later neural responses during pursuit initiation were explained by the integral of retinal velocity errors.

## Results

We instructed the participants to perform a smooth pursuit eye-movement task using random dot kinematograms[10] under different directional cues (Fig. 1a, see Methods for details). Eye positions and velocities were collected using an infrared video camera, and EEG activity was recorded using 64-channel active electrodes.

**Reliability-weighted information integration observed in the smooth pursuit eye movements.** In our study, we independently controlled the expectation of the motion direction and the strength of the sensory stimulus to control the reliability of each piece of information. During the visually guided smooth pursuit eye movement task (Fig. 1a), we manipulated the validity of

directional cues using a block design to control the participants' expectations of motion direction (valid block vs. invalid block). We also manipulated the reliability of the sensory evidence by randomly changing the motion strength of the pursuit target in each trial by modifying the luminance contrast of the stimulus (100 vs. 12%). In the valid block, the direction of the cue was the same as the direction of the pursuit target; therefore, the prior direction information always matched the direction of the target stimulus. In the invalid block, the direction of the cue was randomized, and in 2/3 of the scenarios, it did not match the direction of the pursuit target. We predicted that when the contrast of the stimulus was high, the relative reliability of the target motion information would be significantly higher than that of the cue information. Therefore, the effect of the directional cue on participants' pursuit traces would be minimal. However, when the contrast of the stimulus was low, the relative reliability of the cue motion would be higher than when the contrast of the stimulus was high; therefore, the participants' eye traces would be more affected by the direction of the cue and be attracted towards it.

Smooth pursuit eye movement comprises two stages: open-loop period and closed-loop or steady-state period. The open-loop period of smooth pursuit is known to reflect the feedforward sensory-motor processes (up to 100 ms after the pursuit initiation)[16], in which eye movements were guided solely by sensory inputs. The closed-loop or steady-state pursuit period denoted the duration after the open-loop when the feedback signal for the eye's current position relative to the visual motion target arrived at the pursuit system. During this period, the pursuit system kept correcting and adjusting eye positions to maintain the eyes on the moving visual stimulus. In the behavioral results, we used the open-loop period of smooth pursuit to observe the effect of the sensory cue in the absence of feedback signals. To analyze the open-loop period of smooth pursuit, we visually screened all eye velocity traces from all individual trials and discarded trials with saccadic eye movements in a time window from −100 to 250 ms relative to the visual stimulus onset (Fig. 1b). The eye movement traces of the participants for the same pursuit target stimulus switched depending on the relative strength of the cue and the pursuit target. Figure 1c, d show the mean eye velocity traces of the 14 participants in a time window between −100 and 100 ms from the pursuit latency (timing of the eyes start tracing the stimulus, Supplementary Fig. 1a, b). In the invalid block, the pursuit velocity traces of the participants were notably attracted toward the direction of the cue as if the direction information in the cue and target were combined: as the directions of the cue were randomly selected from the three directions (0°, −30°, −60°), eye velocity traces for the two outer stimulus directions were attracted inward towards the center (−30°) and the traces for the center direction did not change (Fig. 1c, d).

The attraction was stronger when the difference between the cue direction and the target direction was larger (Supplementary Fig 2a, b, solid blue lines vs. solid green and red lines) and when the luminance contrast of the stimulus was low (12%, Fig. 1c, d, blue dashed and red dashed lines). When the cue direction was the same as the pursuit target direction (valid block), the participants' eye velocity traces properly matched the target traces regardless of the contrast of the pursuit target (Fig. 1c, solid blue lines; Fig. 1d, solid red lines). To estimate these results quantitatively, we rotated each eye velocity trace by −30° and calculated the distance between the upper and lower traces (Fig. 1e, Δy1 and Δy2). We then compared the differences in distance between the valid and invalid blocks (Fig. 1f, Δy2 − Δy1). The cue effect was more significant at a low contrast than at a high contrast, quantified by the across-block comparisons of the

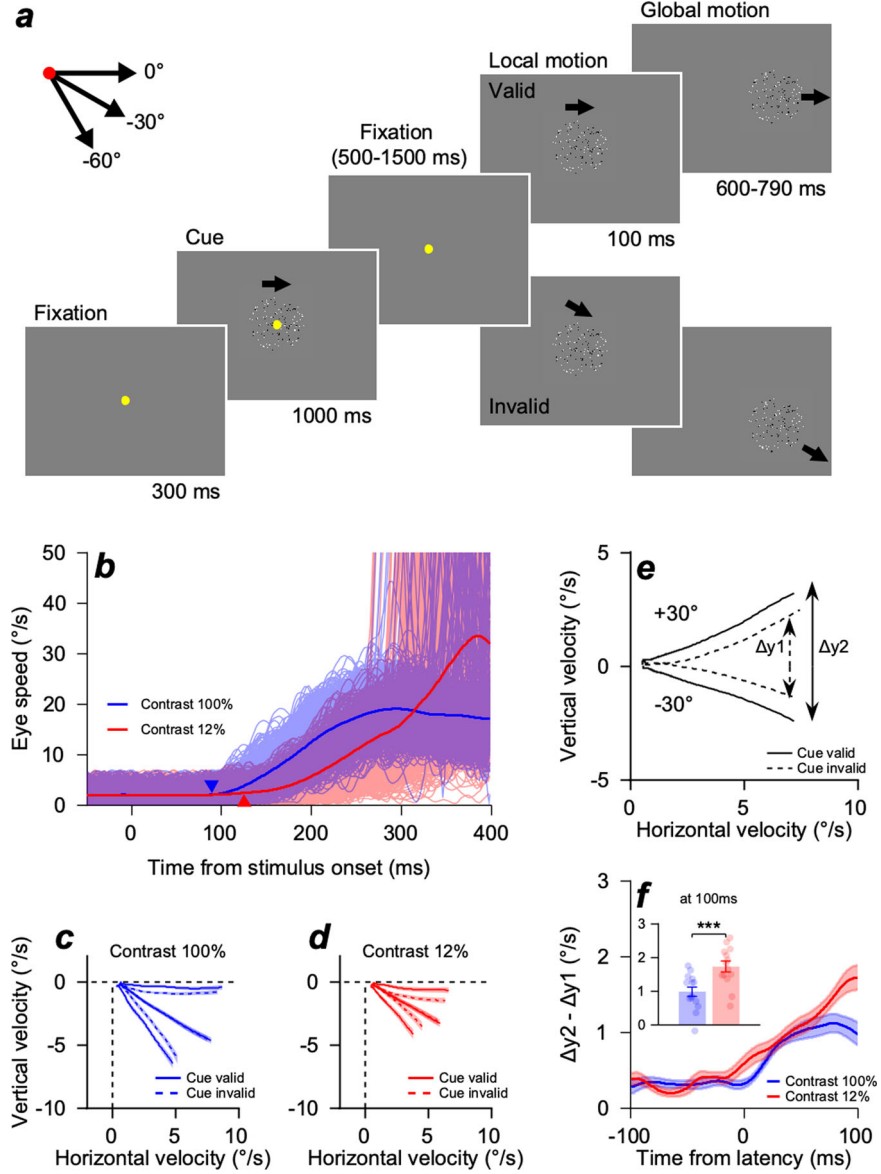

**Fig. 1 Schematics of the smooth pursuit eye movement task design and effect of experimental control on the eye movement traces. a** Each trial began with the onset of a yellow fixation dot at the center of the screen (300 ms). A random dot kinematogram, used as a directional cue, was then presented, and dots (black and white) inside the invisible circular window moved into one of three predetermined directions (0°, −30°, and −60°) at a speed of 16°/s for 1000 ms, whereas the invisible window did not move (pattern motion). After the random dot kinematogram disappeared, the yellow fixation dot remained for the random fixation duration (500–1500 ms). The fixation duration was randomly selected to prevent participants from anticipating the timing of the stimulus onset. After the fixation dot disappeared, the random dot kinematogram, used as a stimulus, reappeared at the center of the screen for 100 ms (local motion) with a pattern motion, followed by the window motion at the same speed and direction at which the dots and window moved together for 600–790 ms (global motion). In a valid block, the direction of a cue and stimulus was identical for all trials. In an invalid block, the direction of a cue and stimulus was randomized within three predetermined directions for all trials. **b** Example eye speed traces from valid blocks of a single participant (JJH). Saccadic eye movements that occurred in a time window between −100 and 250 ms from stimulus onset were discarded from the data. Light-colored traces indicate the eye speed of individual trials, and a dark-colored trace indicates the average eye speed. Each triangle indicates the average timing of the eyes to begin tracing the stimulus (latency) for each contrast condition. **c, d** Average eye velocity traces of 14 participants between −100 and 100 ms from pursuit latency when the target contrast was 100% (Fig. 1c) and 12% (Fig. 1d). The solid lines show the valid trials, and the dashed lines show the invalid trials. The color-shaded areas indicate standard errors. **e** Example of rotated, average eye velocity traces of upper and lower motion directions in Fig. 2c. Each eye velocity trace was rotated by −30° to calculate the distance between the upper and lower traces. y1 indicates the distance between the upper and lower traces of invalid blocks, and y2 indicates the distance between the upper and lower traces of valid blocks. **f** Differences in the eye velocity trace distance between valid and invalid blocks (y2–y1) between −100 and 100 ms from pursuit latency. The bar graphs on the top left show the distance differences at 100 ms after latency within the open-loop period of the smooth pursuit. The color-shaded areas and error bars show the standard errors. ∗∗∗$p<0.001$, two-sample $t$-test.

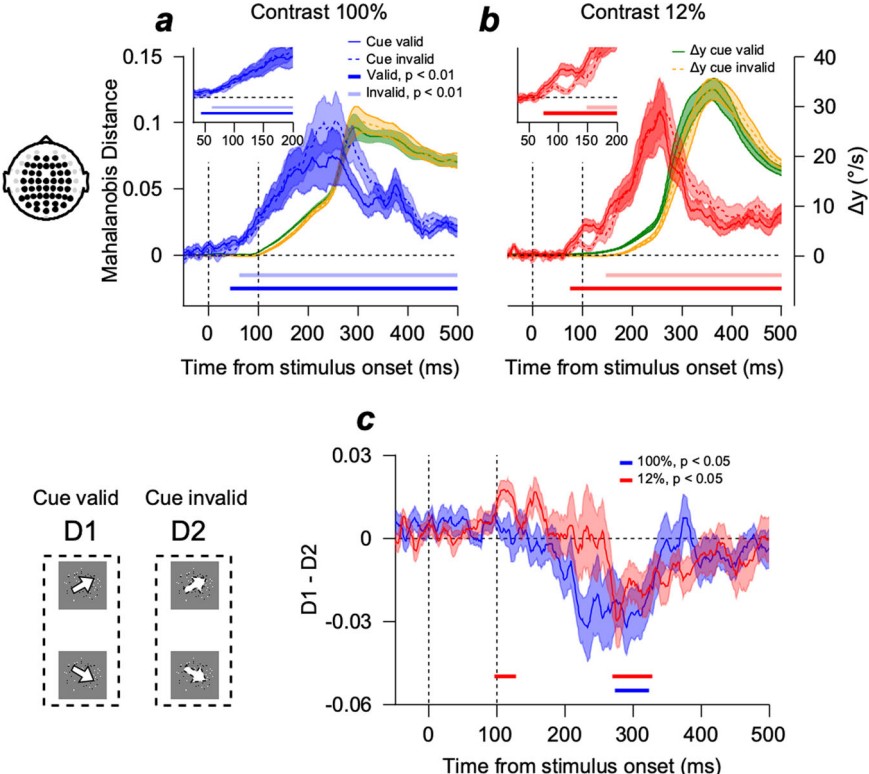

**Fig. 2 Multivariate EEG activity pattern dissimilarity between upper and lower tracking trials estimated using Mahalanobis distance. a** Pattern dissimilarity (Mahalanobis distance) of EEG activity aligned by the stimulus onset when the stimulus contrast was 100%. The solid blue and dashed lines are the EEG dissimilarity when the cue was valid and invalid, respectively. A visual stimulus appeared at 0 ms, and global motion occurred at 100 ms. The solid green line (cue-valid condition) and dashed yellow line (cue-invalid condition) show the eye velocity trace distance differences ($\triangle y$) from Fig. 1e. The color-shaded areas denote standard errors. A figure at the left illustrates the electrodes used for the analysis. Insets show enlarged EEG dissimilarities. The dark-colored line (valid) and light-colored line (invalid) at the bottom of each plot show the time points where Mahalanobis distance was significantly different from zero (two-sided cluster-based permutation test, $n = 14$, cluster-defining threshold $p < 0.01$, corrected significance level $p < 0.01$, 50,000 permutations). **b** Pattern dissimilarity of EEG activity when the stimulus contrast was 12%. The figure format is the same as in Fig. 2a. **c** Difference between EEG pattern dissimilarities of valid and invalid conditions. Neural direction discrimination in the cue-valid condition was significantly better than the discrimination in the cue-invalid condition only when stimulus contrast was 12% (96–128 ms). Later, neural direction discrimination was better in the cue-invalid condition than in the cue-valid condition across both stimulus conditions (two-sided cluster-based permutation test, $n = 14$, cluster-defining threshold $p < 0.05$, corrected significance level $p < 0.05$, 50000 permutations). The color-shaded areas denote standard errors. The illustration at the left shows the pursuit target direction conditions used for calculating D1 and D2 in Fig. 2c.

distance difference (Fig. 1f, red for 12% contrast, blue for 100% contrast). The average distance difference near the end of the open-loop period (at 100 ms after pursuit latency[16]) was the largest (Fig. 1f, left top inset figure, two-sample $t$-test, $n = 14$, $p = 2.25 \times 10^{-5}$). The results suggest that manipulating the relative reliabilities of the cue and pursuit target induced the expected behavioral changes indicating reliability-weighted integration of direction information.

**Decoding the integration of motion direction information from multivariate EEG activity**. Next, we investigated how the reliability-weighted integration of motion information was represented in EEG activity. We compared the multivariate EEG response patterns to two different motion directions (upper direction 0° and lower direction −60°) using the Mahalanobis distances (see Methods for details). The Mahalanobis distance showed the distance between the patterns of EEG activity defined in neural space composed of the number of recording channels[17,18]. This measure considers covariance among channels when calculating the distance, providing variance-normalized neural dissimilarity, which has been proven to be a good measure in showing neural pattern dissimilarities from previous

studies[19–21]. We performed the analysis under different cue conditions (valid vs. invalid cues) and motion strength conditions (100 vs. 12% contrast). Figure 2a, b show the average Mahalanobis distances of the 14 participants. Neural direction discrimination (estimated from the Mahalanobis distance) depended on the contrast of the pursuit target and the congruency between the cue and pursuit target directions (Supplementary Fig. 2c, d). Additionally, neural direction discrimination depended on the motion direction differences, showing smaller Mahalanobis distances when comparing the central (−30°) direction and upper/lower (0°/−60°) directions (Supplementary Fig. 3). The neural direction discrimination (Fig. 2a, b, blue solid, blue dashed, red solid, and red dashed lines) occurred ahead of the direction discrimination estimated from eye movements ($\triangle y$, Fig. 2a, b, green solid and yellow dashed lines).

Significant direction discrimination was faster when the target contrast was high than when it was low. It was faster when the cue direction matched with the target direction than when the cue direction was random (two-sided cluster-based permutation test, $n = 14$, $p < 0.01$, 44 vs. 62 ms for valid and invalid conditions in the high-contrast condition, Fig. 2a, solid blue line vs. blue dashed line; 75 vs. 148 ms for valid and invalid conditions in the low-contrast condition, Fig. 2b, solid red line vs. red dashed line).

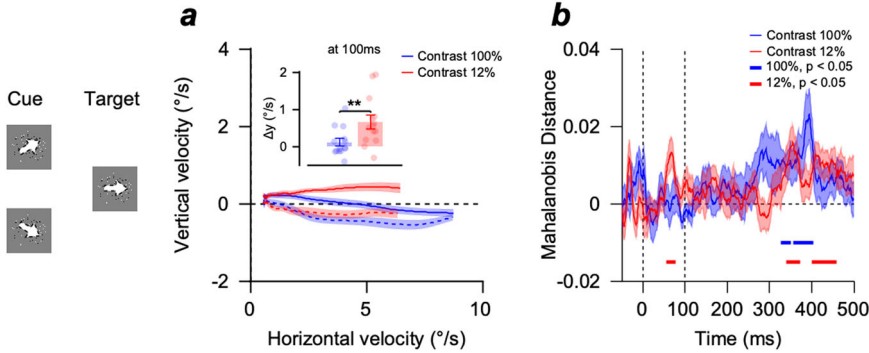

**Fig. 3 Effect of experimental control on the eye movement traces and neural representations during central tracking trials. a** Average eye velocity traces of central target tracking trials ($-30°$) with an upper directional cue ($0°$, colored solid lines) and lower directional cue ($-60°$, colored dashed lines) rotated by $-30°$. The bar graphs in the inset show the distance between two eye velocity traces at 100 ms after pursuit latency for 100% (blue) and 12% (red) contrast conditions. The color-shaded areas and error bars show the standard errors. $**p<0.01$, two-sample $t$-test. The illustration at the left shows motion directions in cue and pursuit targets for calculating eye velocities (Fig. 3a) and EEG dissimilarity (Fig. 3b). **b** EEG activity pattern dissimilarity between central target tracking trials with upper and lower directional cues. A visual stimulus appeared at 0 ms, and global motion occurred at 100 ms. The blue line (100% contrast) and the red line (12% contrast) at the bottom show the time points where Mahalanobis distance was significantly different from zero (two-sided cluster-based permutation test, $n = 14$, cluster-defining threshold $p < 0.05$, corrected significance level $p < 0.05$, 50000 permutations). The color-shaded areas denote the standard errors.

These results indicated that neural direction discrimination of the two motion stimuli was faster when the sensory inputs were strong. Additionally, the initial neural direction responses were modulated by the direction of the cue, suggesting the integration of motion direction information in the cue and pursuit target. These results indicated that the integration of motion information in the cue and target, observed from EEG activity patterns, was weighted by the reliability of each piece of information (Bayesian integration). For a more direct comparison, we tested whether neural direction discrimination in the cue-valid condition was significantly better than that in the cue-invalid condition by calculating the difference between the two Mahalanobis distances (Fig. 2c). Neural direction discrimination was significantly better in the cue-valid than in the cue-invalid condition only when the pursuit target contrast was low (from 96 to 128 ms, two-sided cluster-based permutation test, $n = 14$, $p < 0.05$), which occurred ahead of the pursuit initiation (average pursuit latencies in low-contrast condition were $124.15 \pm 3.48$ ms and $127.47 \pm 3.95$ ms for the cue-valid and -invalid conditions, respectively). Counter-intuitively, neural direction discrimination was worse in the cue-valid than in the cue-invalid condition in both contrast conditions (from 273 to 323 ms for the high-contrast condition, from 269 to 328 ms for the low-contrast condition, two-sided cluster-based permutation test, $n = 14$, $p < 0.05$) after the eyes began moving and around the timing of corrective saccades (at the end of the open-loop period). During this period, neural direction discrimination improved when the direction difference between the cue and the target was larger (Supplementary Fig. 2c, d). The reason for these counterintuitive results is addressed in a later section of this paper.

To further evaluate whether reliability-weighted integration of direction information occurred, we compared the EEG activity patterns and eye velocities to the identical directions of pursuit targets in different cue directions in the cue-invalid condition (comparison between the responses to $-30°$ target direction when cue direction was $0°$ and $-60°$). One of the predictions of reliability-weighted information integration is that the effect of the cue direction on the behavioral and EEG responses is greater when the pursuit target contrast is low (i.e., when the relative reliability of direction information in the cue is higher). Consistent with this prediction, the directions of pursuit eye movements were modulated more by the motion direction

information of the cue when the target stimulus contrast was low (Fig. 3a). The average distance between eye velocity traces between the two cue direction conditions was greater in the 12% contrast condition than in the 100% contrast condition (Fig. 3a, inset, two-sample $t$-test, $n = 14$, $p = 0.0026$). The average Mahalanobis distance between multivariate EEG response patterns to the central target directions in the two cue direction conditions was only significant when the relative reliability of the target stimulus was low (in 12% contrast, from 55 to 77 ms after the stimulus onset, Fig. 3b, two-sided cluster-based permutation test, $n = 14$, $p < 0.05$). We also observed time clusters where the Mahalanobis distances were significant in both target contrast conditions, but they appeared after the eye began to move, where eye movements could influence neural activity.

We then investigated if the neural implementation of Bayesian integration of cue and pursuit target in smooth pursuit is limited to a specific region in the brain. We compared the EEG activity dissimilarity patterns of upper ($0°$) and lower ($-60°$) tracking trials (methods analogous to Fig. 2) in different brain regions by dividing the electrodes according to their locations along the anterior-posterior axis (Fig. 4 and Supplementary Fig. 4). The neural direction discrimination estimated from frontal-central electrodes (Fig. 4a, from 80 to 125 ms, two-sided cluster-based permutation test, $n = 14$, $p < 0.05$) and central-parietal electrodes (Fig. 4d, from 96 to 123 ms, two-sided cluster-based permutation test, $n = 14$, $p < 0.05$) was significantly better in the cue-valid than in the cue-invalid condition when the pursuit target contrast was low. Accordingly, in these brain regions, significant direction discrimination was faster in the cue-valid than in the cue-invalid condition and faster when the target contrast was high (two-sided cluster-based permutation test, $n = 14$, $p < 0.01$, for valid and invalid conditions in the high-contrast condition, 69 vs. 90 ms, Fig. 4b, 92 vs. 97 ms, Fig. 4e, solid blue line vs. blue dashed line; for valid and invalid conditions in the low-contrast condition, 81 vs. 151 ms, Fig. 4c, 134 vs. 167 ms, Fig. 4f, solid red line vs. red dashed line). The results suggest that the Bayesian integration may be primarily implemented in the portion of parietal areas. We could not find the significant effects from the other brain regions (Supplementary Fig. 4).

**Dissimilarity in integrated retinal velocity error signal was predictive of the size of the EEG pattern dissimilarity.** The

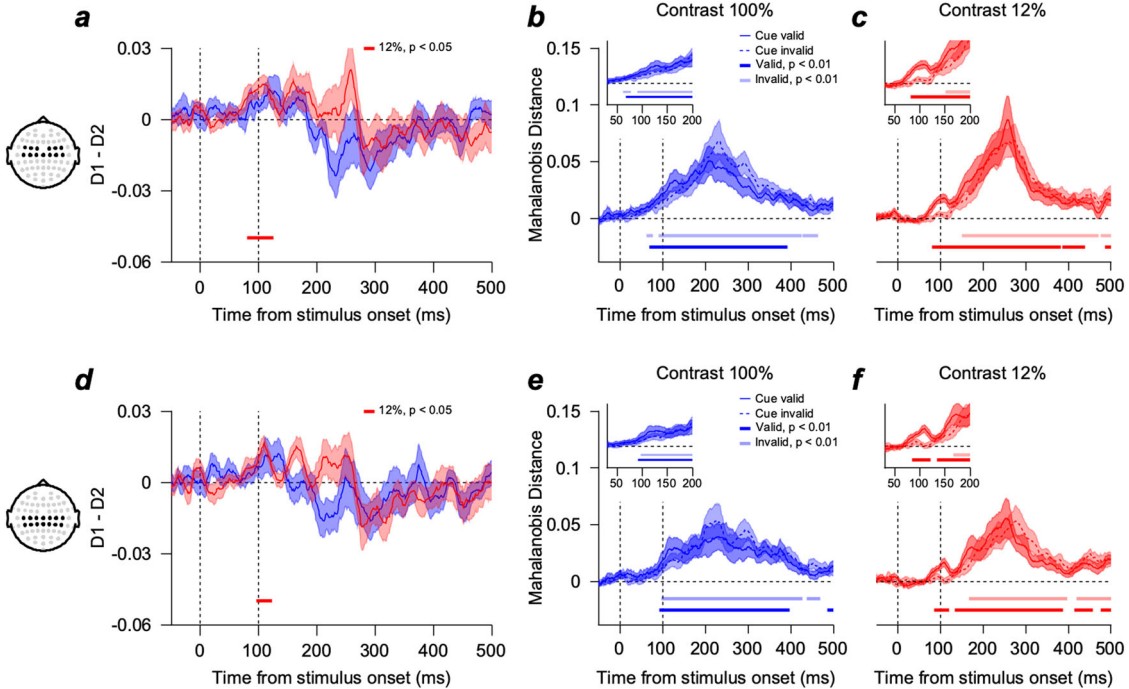

**Fig. 4 Multivariate EEG activity pattern dissimilarity (Mahalanobis distance) between upper and lower tracking trials from frontal-central channels (FC1, FC2, FC3, FC4, FC5, FC6, Cz, C1, C2, C3, C4, C5, and C6) and central-parietal channels (Cz, C1, C2, C3, C4, C5, C6, CPz, CP1, CP2, CP3, CP4, CP5, and CP6).** The figure format for **a–c** and **d–f** is the same as in Fig. 2. **a** Difference between EEG pattern dissimilarities of valid and invalid conditions from the frontal-central channels. Neural direction discrimination was significantly better in the cue-valid than in the cue-invalid condition (two-sided cluster-based permutation test, $n = 14$, cluster-defining threshold $p < 0.05$, corrected significance level $p < 0.05$, 50,000 permutations). **b, c** EEG pattern dissimilarity from the frontal-central channels when the target contrast was 100% (blue) and when the target contrast was 12% (red). The dark-colored line (valid) and light-colored line (invalid) at the bottom of each plot show the time points where Mahalanobis distance was significantly different from zero (two-sided cluster-based permutation test, $n = 14$, cluster-defining threshold $p < 0.01$, corrected significance level $p < 0.01$, 50,000 permutations). **d** Difference between EEG pattern dissimilarities of valid and invalid conditions from the central-parietal channels. The figure format is the same as in Fig. 4a. **e, f** EEG pattern dissimilarity from the central-parietal channels. The format is the same as in Fig. 4b, c. The color-shaded areas denote the standard errors.

Mahalanobis distances estimated using the multivariate EEG activity exhibited faster direction discrimination in high-contrast and cue-valid conditions (Fig. 2a, b). However, the peak amplitudes of the Mahalanobis distances were higher in the low-contrast and cue-invalid conditions than in the high-contrast and cue-valid conditions (Fig. 2a–c). This result was counterintuitive, as a stronger sensory input would result in better motion discrimination. Two broad potential sources contribute to neural direction discrimination. The first is the retinal motion error (relative velocity of sensory stimulus to eye movement), which would work as sensory input to the system. Since there were eye movements, the retinal input (relative to the eye movements) for the motion would continue to change and contribute to the temporal dynamics of the neural signal. The second is motor behavior. Eye movement itself can influence the multivariate pattern of EEG activity and contribute to neural direction discrimination. Additionally, this motor behavior would work as an output of the system. To elucidate the source, we estimated the direction discrimination in motor behavior and sensory input using Mahalanobis distance (Fig. 5, see Methods for details). We calculated the Mahalanobis distances between the two pursuit target direction conditions (0° and −60°, Fig. 5a, b) for the eye velocities and positions as motor behaviors. Direction discrimination in motor behaviors was better and faster in high-contrast, cue-valid conditions than in low-contrast, cue-invalid conditions. Therefore, it is unlikely that these would be the source of the observed neural direction discrimination (Fig. 2a, b). We calculated the retinal velocity error (sensory motion velocity vector – eye velocity vector) and estimated the difference in the

sensory velocity error between the two motion direction conditions using the Mahalanobis distance (Fig. 5c). Because these retinal velocity errors should be integrated over time to initiate eye movements, we calculated the Mahalanobis distance of the integrals of retinal velocity error (IRVEs) for the two direction conditions (Fig. 5d). Interestingly, we observed that direction discrimination estimated using the IRVEs exhibited a similar pattern to neural direction discrimination: they were better in the low-contrast, cue-invalid condition than in the high-contrast, cue-valid condition.

To quantitatively test whether this intuition was held in individual participants and elucidate which sensory input and motor output components contributed to the neural responses, we modeled the EEG pattern dissimilarity with the weighted linear summation of sensory and motor dissimilarities (Fig. 6a and Supplementary Fig. 5, see Methods). In this initial full model, we used eye position and eye velocity as the motor behaviors and the IRVE and the retinal velocity error as the sensory inputs. We used the same set of parameters (six free parameters: weights for eye velocity, eye position, retinal velocity error, and the IRVE, timing parameter for sensory input $t_s$, and timing parameter for motor output $t_m$) for all four conditions (cue validity and stimulus contrast) to explain the variation of temporal response patterns of EEG dissimilarity across conditions. This model explained 68.9% of the EEG dissimilarity variance (68.9 ± 13%). The estimated weight for the IRVE was dominant in elucidating the neural data (Supplementary Fig. 5). To simplify the model, we only used eye velocity as the motor behavior and the IRVE as the sensory input (four free parameters, Fig. 6) in the linear

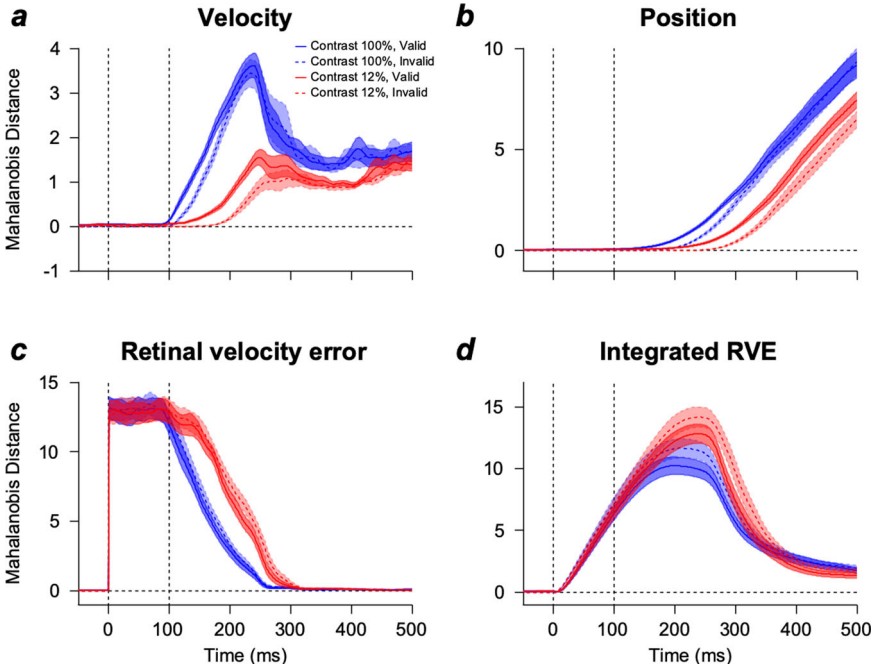

**Fig. 5 Pattern dissimilarity across the horizontal and vertical components of the motor and sensory features of eye movements. a–d** Pattern dissimilarity (Mahalanobis distance) across the horizontal and vertical components between two motion direction conditions for **a** eye velocity, **b** eye position, **c** retinal velocity error (velocity of sensory motion – eye velocity), and **d** integral of retinal velocity error. The color-shaded areas indicate the standard errors.

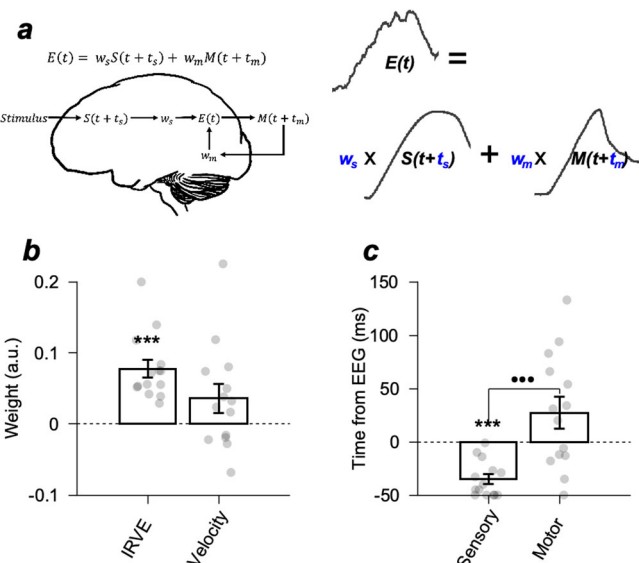

**Fig. 6 Description of the linear model and estimated parameters.**
**a** Modeled EEG activity pattern dissimilarity for the two motion directions with the weighted linear summation of sensory input and motor output differences between the two direction conditions. The IRVE was used as the sensory input, and eye velocity was used as the motor output (behavior). $E(t)$ is the EEG pattern dissimilarity, $w_s$ is the weight for the sensory input dissimilarity, $S(t)$ is the integrated retinal velocity error dissimilarity, $t_s$ is the relative time of the sensory input, $M(t)$ is the eye velocity dissimilarity, $w_m$ is the weight for the motor output dissimilarity, and $t_m$ is the relative time of the motor output. **b** Averages of estimated weights for the IRVE and eye velocity. **c** Averages of estimated times relative to EEG time for sensory input and motor output. The error bars denote the standard errors. *** $p<0.001$, one-sample $t$-test; ••• $p<0.001$, two-sample $t$-test.

summation. This reduced model explained 63.8% of EEG dissimilarity variance on average (63.77 ± 13.29% SD), which still explained the EEG dissimilarity pattern. The estimated parameters are shown in Fig. 6b, c. The weights for the IRVEs were significantly higher than zero (one-sample $t$-test, $p = 2.699 \times 10^{-5}$). In contrast, the weights for eye velocities were not significant (one-sample $t$-test, $p = 0.089$), consistent with our intuition.

The estimated timing for the eye velocity was after the EEG dissimilarity ($t_m = 27.21 \pm 14.84$ ms, one-sample $t$-test, $p = 0.079$), whereas the estimated timing for the IRVE was in advance of the timing of EEG dissimilarity ($t_s = -35.29 \pm 4.66$ ms, one-sample $t$-test, $p = 2.739 \times 10^{-6}$) and was significantly ahead of the timing for the eye velocity (two-sample $t$-test, $p = 3.006 \times 10^{-4}$). Therefore, IRVE dissimilarity was predictive of EEG dissimilarity, and EEG dissimilarity was weakly predictive of eye velocity dissimilarity. This temporal relationship was the same in the full model with all components considered. The estimated timing for the sensory inputs (retinal velocity error and IRVE, $t_s$) was in advance of the EEG timing. The estimated timing for the motor behavior (eye velocity and position, $t_m$) followed the EEG (Supplementary Fig. 5). In the even reduced model, one might argue that the usage of IRVE and the eye velocity is redundant. Although IRVE and eye velocity have common temporal components (eye velocity trace), they worked as independent elements when estimating the linear model. The weights of the IRVE and velocity assigned to the individual templates were not correlated across participants ($r = -0.1033$, $p = 0.727$, Supplementary Fig. 6). This indicates that each element played different roles in explaining the EEG dissimilarities.

Lastly, we used the modeling approach to test whether the simple retinal input and motor output could explain the cue-induced neural modulation we observed. If the neural modulation we observed resulted from the Bayesian integration, the effect should not be simply explained by either the sensory input or the motor output. We removed the sensory input and motor output

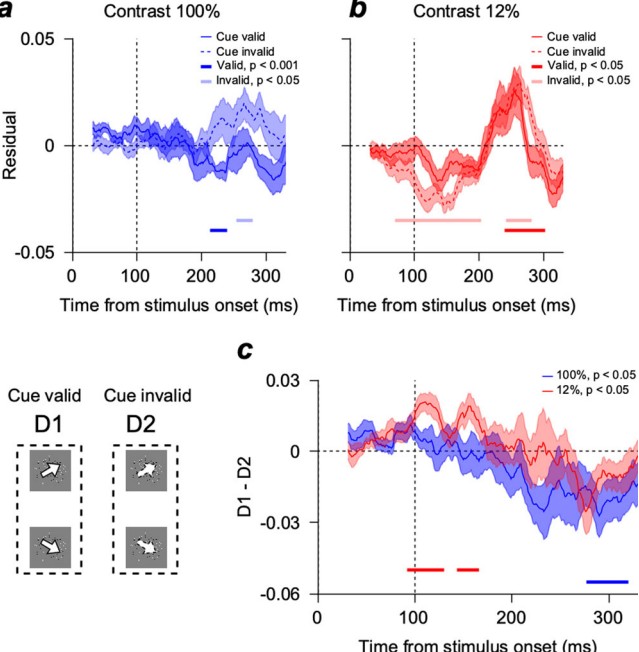

**Fig. 7 Residuals from the model prediction. a** EEG activity pattern dissimilarity (Mahalanobis distance) of the residuals in the time window of 30 and 330 ms from the stimulus onset when the stimulus contrast was 100%. The solid blue and dashed lines are the EEG dissimilarity when the cue was valid and invalid. **b** Pattern dissimilarity of the residuals when the stimulus contrast was 12%. The figure format is the same as Fig. 7a. The dark-colored line (valid) and light-colored line (invalid) at the bottom of each plot show the time points where Mahalanobis distance was significantly different from zero (two-sided cluster-based permutation test, $n = 14$, cluster-defining threshold $p < 0.05$, corrected significance level $p < 0.05$, 50,000 permutations). **c** Difference between the residual dissimilarity patterns of valid and invalid conditions. Neural direction discrimination in the cue-valid was significantly better than in the cue-invalid condition when stimulus contrast was 12% (92–130 ms, two-sided cluster-based permutation test, $n = 14$, cluster-defining threshold $p < 0.05$, corrected significance level $p < 0.05$, 50,000 permutations). The color-shaded areas denote standard errors.

components by subtracting the model prediction from the EEG dissimilarity pattern (Fig. 7a, b). Therefore, we expect the remaining EEG activity pattern to contain components not accounted for by the pursuit system's simple sensory input and motor output. When the stimulus contrast was high, the model explained the EEG dissimilarity before pursuit initiation (~100 ms after the target onset), regardless of the validity of the cue (Fig. 7a). However, when the stimulus contrast was low, model prediction overestimated the early neural direction discrimination (Fig. 7b). The overestimation occurred because we did not consider the strength of the motion information in the model building. In both contrast conditions, retinal input components of the model only depended on the speed and direction of the pursuit target. Therefore, the EEG motion discriminability would be weaker than the model prediction when the stimulus contrast was low. An important aspect of the data here is that the overestimation was larger in the cue-invalid than the cue-valid condition (Fig. 7b), showing the neural direction discrimination was better in the cue-valid than in the cue-invalid condition. Comparison between the cue-valid and cue-invalid conditions revealed significant differences when pursuit target contrast was low (Fig. 7c, from 92 to 130 ms, two-sided cluster-based permutation test, $n = 14$, $p < 0.05$). This demonstrates that neural

direction discriminability was weaker in the cue-invalid than in the cue-valid condition only when the sensory evidence was weak. These results strongly suggest that the early neural modulation reflected the neural process of reliability-based information integration, which could not be simply explained by sensory input or motor output.

## Discussion

This study examined the reliability-based optimal information integration process revealed in the behavior and EEG activity during sensory-guided oculomotor behavior. Consistent with our prediction based on Bayesian inference, we demonstrated that participants' behaviors and multivariate EEG activity patterns were more affected by the motion cue when the motion strength of the pursuit target was weak. In addition, we observed that multivariate EEG activity represented the retinal motion information most strongly and explained the variability in the EEG activity pattern. The multivariate EEG activity pattern was sufficiently sensitive to reveal neural evidence of reliability-based information integration: neural dissimilarities responding to the sensory stimuli before the eye started moving exhibited small but significant Bayesian modulations. The temporal response pattern of the neural dissimilarity in open-loop smooth pursuit represented the IRVEs. Therefore, motion information represented in the multivariate EEG activity during sensory-motor behavior contains both the evidence of Bayesian integration and the Galilean relative velocity of sensory motion.

Under the Bayesian inference framework, the brain optimizes behavior by performing a weighted sum of information based on its reliability[7,8,10,22–24]. The weight of the higher reliability input has a higher value, encoded in the amplitude of the neural responses[25]. Recent EEG studies on this theme revealed that neural activity over the sensorimotor cortices might have a functional role in Bayesian inference[26,27]. Under the motor adaptation paradigm, post-movement beta synchronization is negatively correlated with previous errors[28,29], and the beta band activity may reflect the precision estimates of motor prediction[28]. Palmer et al.[30] demonstrated that sensorimotor beta oscillations are correlated with changes in precision weighting parameters. Adams et al[14]. modeled steady-state smooth pursuit to track a sinusoidally oscillating target using an active inference framework. They demonstrated that the gain control of V1 neural activity is linked to the control of the subjective precision of sensory information[15]. These findings suggest that the Bayesian framework provides plausible insights into how the human brain optimizes motor behavior. However, studies investigating how optimal integration of multiple pieces of information under dynamically changing conditions are represented in neural activity and mediate behavior in humans are scarce. In this paper, utilizing the high temporal resolution of EEG and visually guided smooth pursuit eye movements composed of initial open-loop (feedforward) and steady-state (feedback) periods[16,31], we report the dynamics of the reliability-based information integration revealed through multivariate EEG activity patterns.

Corresponding with the results of studies on cue integration in humans through Bayesian coding[32–35], we have shown that the integration of prior knowledge for motion direction with sensory motion information follows the prediction of this Bayesian inference in the smooth pursuit eye movements of a human. The effects of cue direction on both pursuit behavior and EEG representation of motion direction were more pronounced when the contrast of the pursuit target was low, which effectively decreased the relative reliability of sensory input when the cue and pursuit target were integrated (Figs. 1, 2). The pursuit behaviors were more attracted toward the cue direction with the

corresponding changes in EEG representation of motion direction when the difference between the cue direction and the target direction was larger, which is consistent with the prediction of the Bayesian inference (Supplementary Fig. 2). This was demonstrated when we evaluated the effects of different cue directions on the same pursuit target direction. Behavioral responses to the identical pursuit target were modulated using the direction of the cue. This modulation was stronger when the relative reliability of the cue was higher than that of the pursuit target (Fig. 3a). The different cue directions made the multivariate EEG response patterns to the identical pursuit target different only when the contrast of the pursuit target was low (Fig. 3b). These results provide an interpretation of how the dynamics of the information integration process represented in the EEG pattern mediates movement control in fast-changing environments.

Studies demonstrated the implementation of Bayesian inference in the cerebral cortex[36–38]. Funamizu et al.[39] found the features of dynamic Bayesian inference from the posterior parietal cortex (PPC) and adjacent posteromedial cortex (PM). The macaque perceptual decision-making study showed that the responses of neurons in the lateral intraparietal area (LIP) reflected substantial temporal integration of visual motion signals coming from neural activity in the middle temporal (MT) area toward the choice of decision target[40]. Therefore, LIP could be a candidate area for Bayesian information integration. Sensory motion information in the area MT also projects to the medial superior temporal area (MST). Neurons in MST integrate sensory signals to compute optic flows[41,42]. Several studies showed that the Bayesian integration of vestibular and visual motion signals happened in single neural MST activities[22,42,43]. These results reported in MST suggested that Bayesian information integration might also occur in MST neural activity.

In smooth pursuit eye movements, neural activity in the area of MT provides sensory motion signals to the oculomotor system, and the sensory information guides the motor behaviors. Previous studies showed that single neural activities in the area MT were correlated with the variation of pursuit speed[44], direction[45], and latency[46]. Previous studies suggested that the gain of this sensory motion information is modulated by neural activity in the frontal eye field smooth eye movement region (FEF_SEM)[7,8,10,45], implicating the role of FEF_SEM as a source for the prior expectation. Collectively, the results of these studies suggest that the initiation of smooth pursuit eye movements is possibly driven by the gain-modulated (expectation-modulated) sensory motion information represented in frontoparietal and posterior parietal areas.

In our results, the neural direction discrimination before the initiation of the pursuit was significantly better in the cue-valid condition only when the contrast was low, which was intuitively expected as the result of the Bayesian inference (Fig. 2c). We confirmed that this was not driven simply by the sensory input or motor output variation (Fig. 7c). Moreover, we found the effect was maximally represented in the frontoparietal and posterior parietal areas where electrodes could detect neural signals from the MT, LIP, MST, PPC, and FEF areas (Fig. 4a, d). These results suggest that the EEG dissimilarity pattern we observed before the initiation of the pursuit contains information on the neural modulation under the Bayesian framework with extra-retinal origin.

Previous studies suggest that sensory information integration drives the initiation of smooth pursuit eye movements. A few studies have investigated the neural representation of motion during smooth pursuit eye movements in humans[15,47,48]. A recent study in our laboratory investigated neural representations using multivariate analysis of EEG activity[48]. Using multivariate analysis, we successfully extracted motion direction information and demonstrated the temporal dynamics of the representation,

which was consistent with the current knowledge of information processes in the smooth pursuit of eye movements. We also observed that this motion information variation is tightly correlated with behavioral variation[47,48]. Although these were informative in understanding the information processes underlying human oculomotor behaviors, we did not know what information content drove the EEG activity during the smooth pursuit eye movements, i.e., whether it was sensory motion information, retinal motion errors, motor preparation, planning, or motor behavior itself. In this study, we tested which information was primarily represented in multivariate EEG activity using multiple stimuli and prior conditions that successfully constrained the linear model we built. The classical model of smooth pursuit eye movements assumes retinal velocity errors as an input and eye velocity signal as an output[49–51]. Therefore, we initially sought to model the EEG pattern dissimilarity with retinal velocity errors as inputs and eye velocity as an output. However, in the brain, eye movement is controlled by position signals generated from brainstem neurons (abducens nucleus)[52]. To this end, the retinal velocity errors should be transformed into the motor command that controls the eye position. Moreover, we do not know what information would be represented in the EEG activity pattern. Therefore, we included the possible sensory inputs (retinal velocity error and integral of retinal velocity error) that could drive the EEG activity and motor outputs (eye velocity and position) that could be explained by or controlled by the EEG activity in the model (Supplementary Fig 5). For each possible input, we estimated the dissimilarity pattern to find that has a similar direction discrimination pattern with the neural direction discrimination pattern (Fig. 5). The model appeared to be dominated by the IRVE, showing significant and dominant weight (Supplementary Fig 5a). Therefore, we reduced the model with fewer components without losing much of the explainability of the model (Fig. 6).

The IRVE may dominate in explaining EEG pattern dissimilarity because the retinal velocity error should accumulate to produce position information to drive the motor behavior. This is conceptually different from the retinal position error. In our experimental design, the pursuit target had a velocity change without any position change (initial 100 ms; see Methods). In addition, the occurrence of visual motion is the main factor that initiates smooth pursuit, unlike saccadic eye movements, in which the displacement of the target position is the main driving factor[16,53]. Another possible explanation for the IRVE is that the EEG activity pattern may represent the integration of the retinal velocity error for the catch-up saccade. At the end of the open-loop smooth pursuit, the oculomotor system integrates the retinal error signal and uses it to correct the disparity between the visual target and eye movement[16,53]. This aspect has been suggested to be included in the smooth pursuit model when the interaction between the smooth pursuit and catch-up saccade happened[54,55]. In our results, the EEG dissimilarity pattern reached the maximum value close to the onset of the catch-up saccade (~250 ms from motion onset, Fig. 1a, b, and Supplementary Fig 2c, d). Therefore, the IRVE may correspond to the motor planning signal for initiating a corrective saccade at the end of the open-loop period.

Another possible explanation is the involvement of cognitive factors. Attention is known to modulate sensory neural activity[56–60]. In human studies, visual-spatial or feature-based attention[61] has been reported to modulate EEG activity. Therefore, participants may have adjusted the gain of sensory information when the sensory evidence was weak[15], probably to compensate for unreliable sensory information. However, in our experiment, it was difficult to explain the results using attentional modulation. Stimulus contrast was randomized such that participants could not predict the contrast of the stimulus and adjust

their attention in such a short period (~300 ms). In addition, attention still cannot explain why neural discrimination is better when direction cues are invalid. If attention improved neural direction discrimination, it would have been better when the cue was valid. Therefore, we conclude that the neural direction discrimination we observed can be best explained by the reliability-weighted integration of motion direction information (for the initial neural responses) and the IRVEs (for the later neural responses).

Previous studies have shown that eye movements can elicit noticeable EEG responses[62]. For example, macrosaccades and microsaccades can evoke EEG gamma-band activity[63,64]. In addition, extraocular muscle activity resulting from eye movement can propagate to the EEG as a spike potential[64]. Therefore, it is crucial to control the effect of eye movements on EEG activity and clarify that eye movements did not drive our results. To remove the potential effect of saccadic eye movements, we discarded trials with any saccadic or microsaccadic eye movements in the time window between -100 and 250 ms from the visual stimulus onset before all analyses. While fixating (from −100 to 100 ms relative to the stimulus onset), we excluded trials with eye speeds exceeding 5°/s. During pursuit (from 100 to 250 ms), any trials considered saccades through visual inspection, with speeds higher than 20°/s, were discarded. In addition, because the latency of the effect of saccadic eye movements to appear on ERP responses was longer than 100 ms[62], microsaccadic or macrosaccadic eye movements had minimal to no effect on EEG responses from 0 ms to ~350 ms relative to stimulus onset. Furthermore, considering the timing of the increase in the dissimilarity pattern of EEG and motor behaviors (velocity and position), it is unlikely that the dissimilarity pattern of EEG that we observed resulted from eye movement.

Moreover, if the EEG responses simply reflected eye movements, the dissimilarity pattern of the EEG would have been more distinctive when the difference between the eye movement traces was more distinguishable. The EEG dissimilarity pattern was noticeably larger in the cue-invalid than in the cue-valid condition close to the end of the open-loop period and in the 12% contrast condition than in the 100% contrast condition. However, the eye velocity difference between the upper and lower traces was larger in the cue-valid than in the cue-invalid condition and in the 100% contrast than in the 12% contrast condition (Fig. 2a, b). These findings suggested that the EEG responses before 350 ms from the stimulus onset reflected the IRVE and were not a byproduct of the eye movements.

In this study, we used the Mahalanobis distance[18] to estimate direction information. Other multivariate analysis methods, such as the inverted encoding model, can be used to extract motion direction information[65]. However, the multiple experimental conditions that we required in this experiment (different contrasts, cue directions, and pursuit target directions) necessitated the usage of a limited number of motion directions, making it difficult to use parametric, model-based approaches. The Mahalanobis distance is a straightforward measure of EEG activity pattern dissimilarity and has been proven successful in identifying orientation information in working memory experiments[19–21]. This enabled us to obtain robust and sensitive measures for discriminating motion directions represented in the multivariate EEG activity and evaluate reliability-weighted information integration. However, the multivariate pattern analysis of EEG activity requires many trials. An insufficient number of trials may result in unreliable noisier neural representations. This limited us to observing the underlying reliability-weighted information integration representation in the multivariate EEG activity at a general level. Despite the limitation, the use of this measure provides essential benefits over univariate or other multivariate

approaches, making it difficult to achieve the current level of sensitivity. In addition, using the Mahalanobis distance, we focused on the direction differences instead of the direction representation itself. Thus, we prevented common components (common temporal dynamics of sensory motion, motor information, or eye movement-related signals) from dominating the EEG activity patterns. It is particularly beneficial in modeling EEG activity patterns using sensory and motor-related components. If we used the EEG activity pattern for each direction—instead of the difference—for linear model building, it would have been challenging to isolate the contributing components because of the common temporal dynamics of EEG activity patterns across the motion directions.

## Methods

**Participants**. We recruited 23 human participants for this experiment. Before each experiment, all participants were informed of the experimental contents, and their written consent was obtained. All experimental protocols were approved by the Institutional Review Board of Sungkyunkwan University. Before each experiment, we verbally informed the participant of the experiment structure, including the trial structure and block design. We conducted behavioral training sessions for an hour before the EEG recording experiment. If more than 30% of the training session trials contained frequent saccadic eye movements during fixation and pursuit, we excluded the participants from the EEG recordings. Six of the 23 participants were excluded. EEG and smooth pursuit eye movement data were collected from the remaining 17 participants with normal and corrected-to-normal vision. The participants visited us three times for EEG recordings to enable us to obtain a sufficient number of trials for each experimental condition for the multivariate analysis. Out of the 17 participants, one was excluded from further analysis because of a low EEG recording quality: more than 50% of independent components (ICs) were detected as artifacts from the ADJUST algorithm used in preprocessing EEG data (see below for details). Data from two participants were excluded because of insufficient trials (data from only 2 days were available). Behavioral and EEG data from the remaining 14 participants were used for further analysis. Each participant conducted an average of 1064 trials per day, and we combined three days' data, which resulted in an average of 3269 trials per person.

**Stimuli and task design**. Figure 1a shows the task design. We performed a simple smooth pursuit eye movement task using a visual cue. Visual stimuli were displayed on a gamma-corrected 20-inch cathode ray tube monitor (Hewlett Packard, p1230). The monitor was positioned 60 cm from the participants, covering 36.9° × 28.1° of the visual field. The monitor had a spatial resolution of 1600 × 1200 pixels and a vertical refresh rate of 85 Hz. All visual stimuli were presented on a gray background (32.7 cd/m²) in grayscale, with a luminance range from 0 to 72.5 cd/m².

We used random dot kinematograms as the visual motion cue and pursuit target. The dot patch comprised equal numbers of randomly mixed black and white dots inside a 4.5° diameter circular patch. The nominal contrast of the dot patch was defined as the difference between the luminance of the black and white dots divided by the sum of their luminances. All the experiments were conducted in a dark room with a display monitor as the primary illumination source. Therefore, the average luminance level at the participant's location in the room during the experiment was 32.7 cd/m² (equal to the measured luminance of the monitor on a gray background).

The task comprised two blocks that depended on the validity of the visual motion cues: valid and invalid blocks. Each trial in the blocks began with a yellow fixation point (0.3° × 0.3°) presented at the center of the screen. After 300 ms, random dot kinematograms appeared at the center of the screen. All the dots in the patch began to move at 16°/s in one of three predetermined directions (0°, −30°, and −60°) with 100% coherence for 1000 ms while the invisible circular aperture of the random dot patch was stationary (a local motion). The contrast of the random dot patch was 100% for all the trials, and the direction of the random dot patch during this phase was used as a cue. The random dot patch then disappeared for a random duration between 500 and 1500 ms. Participants were required to remain fixated at the fixation point within a 2° invisible square window during these periods. Immediately after the extinction of the fixation point, random dot kinematograms reappeared, and a local motion occurred for the initial 100 ms; subsequently, all dots and the aperture moved together with the same speed and direction as the local motion for a random duration between 600 and 790 ms. In the valid block, the direction of the pursuit target was always identical to the direction of the cue. In the invalid block, the direction of the pursuit target was randomly selected from three predetermined directions regardless of the motion direction of the cue. The contrast of the random dot patch during this phase was randomly selected as either 100 or 12%. The participants had to track the random dot patch within a 5° invisible square window around the center of the patch until the target stopped moving and disappeared from the screen. Each block consisted of 72 trials, and valid and invalid blocks were alternately presented. In each

alternation of blocks, we verbally informed the participants of the block changes and told them whether the upcoming block was cue-valid or cue-invalid. We also informed the participants that their eye positions were monitored during each trial, and the trial would be aborted if they failed to maintain their eyes on the target (fixation point and the dot patch) within the given sizes (2° or 5°) of invisible monitoring window centered on the targets. Participants were provided 1000-ms pauses between each trial and a 1-min break between each block.

**Data acquisition**. The horizontal and vertical eye positions and velocities of each participant were recorded using an infrared eye tracker (EyeLink 1000 Plus, SR Research) with a sampling rate of 1 kHz. We displayed the visual stimuli and acquired eye position signals using a real-time data acquisition program (MAES-TRO, https://sites.google.com/a/srscicomp.com/maestro/) used in a previous study. We calibrated the eye position by presenting a 0.3°×0.3° circular white dot at nine positions around the screen ([0°, 0°], [10°, 0°], [10°, 10°], [0°, 10°], [−10°, 10°], [−10°, 0°], [−10°, −10°], [0°, −10°], and [10°, −10°]). We performed a recalibration during the experiment if the eye positions were considerably away from the fixation point during fixation due to a significant amount of head movement. The recalibration of the offset did not affect the results or conclusions of the paper because behavioral data were mainly focused on the relative difference across conditions or temporal changes. We used a custom-built photodiode circuit to record the precise timing of the visual stimuli. A 2° × 2° white square stimulus was turned on at the upper-left corner of the screen where the photodiode was located whenever the target stimulus was turned on.

EEG data were recorded using 64-channel active electrodes (actiCAP, Brain Products, GmbH) and an EEG amplifier (BrainAmp, Brain Products, GmbH) with a sampling rate of 5 kHz. Before the recording, we decreased the impedances of the electrodes to under 5 kΩ by injecting an electrolyte gel (SupverVisc Gel), and the impedance of the electrodes was maintained under 25 kΩ throughout the EEG recording. All behavior control-related information was transmitted to the EEG recording system online using a custom-built hardware interface and a digital input and output device to synchronize the behavioral and EEG data.

**Data preprocessing**. Before analyzing the EEG data, we preprocessed the data to detect and remove artifactual components that originated from faulty electrodes (disconnection and sudden changes in impedance) and participant movements (head and eye movements). All preprocessing procedures were performed in MATLAB (Mathworks, Inc.) using the EEGLab subroutines[66] and FieldTrip toolbox[67]. We first downsampled the EEG data from 5 to 1 kHz. We then applied a detrending method (tenth-order polynomial fit) to remove slow drifts[68]. Next, we removed noisy channels using the artifact subspace reconstruction (ASR) routine[69] and re-referenced all channel data to the average[70]. Line noises (60, 120, and 180 Hz) were removed using the cleanline EEGLab plugin. We applied independent component analysis (ICA)[71] to identify the artifactual components induced by eye movements and blinks. Finally, ICs determined as artifacts were removed from the data using ADJUST EEGLab plugins[72].

**Multivariate pattern analysis of EEG**. We used the Mahalanobis distance[17,18] to compare the multivariate patterns of EEG activity between the two sets of trials. This analysis used only EEG recordings from 47 of the 64 channels, excluding data from electrodes that were prone to muscle activity and noise (Fp1, Fp2, AF7, AF8, F7, F8, FT7, FT8, FT9, FT10, T7, T8, TP7, TP8, TP9, TP10, and Iz). The trials were divided by pursuit directions (0°, −30°, and −60°). In each trial, we calculated the Mahalanobis distance of the trial to the average of all other trials from the same direction condition ($D_{11}$) and the Mahalanobis distance of the trial to the average of all other trials from different direction conditions ($D_{12}$) by performing a leave-one-trial-out cross-validation approach[19]. We estimated the Mahalanobis distance at each time point using the following equations:

$$D_{11} = \frac{1}{n}\sum_{i=1}^{n}\sqrt{\left(TD_1 - td_1^i\right)^T \times pC^+ \times \left(TD_1 - td_1^i\right)} \quad (1)$$

$$D_{12} = \frac{1}{n}\sum_{i=1}^{n}\sqrt{\left(TD_2 - td_1^i\right)^T \times pC^+ \times \left(TD_2 - td_1^i\right)} \quad (2)$$

$TD_1$ and $TD_2$ are the average multivariate activities (excluding the activity from the current trial) in directional conditions 1 and 2 (1× channels), $pC^+$ represents the pseudo-inverse of the pooled covariance matrix (channels× channels), and $td_1^i$ is the activity of the given trial $i$ (1× channels) in direction condition 1. We used the variance stabilization method to pool the covariances from the multivariate EEG activity of the two directional conditions. To prevent any bias in calculating the Mahalanobis distances, we matched the number of trials of the pair by randomly selecting trials from the larger sample size group. We iterated this process ten times and used the average Mahalanobis distances. The dissimilarity of the multivariate EEG activity pattern between the two direction conditions was estimated using the difference between the two Mahalanobis distances, $D_{11}$ and $D_{12}$ ($D_{12} - D_{11}$). When the patterns of EEG activity between the two direction conditions were disparate, $D_{12} - D_{11}$ was higher than zero.

Before this analysis, we smoothed the EEG signal from each electrode using a 20-ms (±10 ms) rectangular time window, with a 1-ms step size, from −100 to

500 ms relative to the pursuit target onset for the analysis of EEG activity during smooth pursuit eye movements. To combine the multiple days' data for each participant, we normalized each day's EEG data by the maximum values across channels, times, and trials. We normalized each channel's data using the mean and standard deviation (SD) across times and trials.

**Multivariate pattern analysis of the eye movements**. To compare eye movements induced motor (eye velocity and eye position) and sensory (retinal velocity error and integrated retinal velocity error) components during the smooth pursuit in the two direction conditions, we calculated the Mahalanobis distance to derive variance-normalized dissimilarity pattern using the horizontal and vertical components of eye movements. The other procedures used to calculate the Mahalanobis distance were the same as the activity pattern analysis of the EEG: using Eqs. 1 and 2, we obtained the difference between the two Mahalanobis distances ($D_{12} - D_{11}$).

**A linear model of multivariate EEG pattern dissimilarity**. To evaluate the relationship among retinal input, sensory, and motor information represented by EEG activity and motor behavior (Fig. 6a), we developed a linear model that explained the multivariate EEG activity from retinal inputs and motor behaviors using the following equation:

$$E(t) = w_s S_{IRVE}\left(t + t_s\right) + w_m M_{vel}\left(t + t_m\right) \quad (3)$$

$E$, $S_{IRVE}$, and $M_{vel}$ denote the dissimilarities in multivariate EEG activity, retinal inputs, and motor behaviors between the two direction conditions, respectively, estimated from the Mahalanobis distances. $E$ is the EEG activity pattern dissimilarity (Fig. 2), $S_{IRVE}$ is the integrated retinal velocity error dissimilarity, and $M_{vel}$ is the eye velocity dissimilarity (Fig. 5a, d). $w_s$ and $w_m$ denote the coefficients of each variable, and $t_s$ and $t_m$ represent the time shifting of the sensory and motor data. The parameters were estimated using the least-squares method (NOMAD algorithm[73]).

Model estimation was performed using the EEG, retinal input, and motor behavior from four different experimental conditions simultaneously (valid target with 100 or 12% contrast conditions and invalid target with 100 or 12% contrast conditions). We used the 300-ms duration of EEG data for the model estimation, from 30 to 330 ms relative to stimulus motion onset. Because we wanted to model the open-loop period of the smooth pursuit eye movements, we considered the initial 300 ms of the time window after the motion onset to include the sensory input to the pursuit system and the effect of the resultant motor behavioral output on the EEG data. We considered a neural response latency of ~30 ms; therefore, the choice of the time window was between 30 and 330 ms. The choice of the EEG time window would not affect the conclusion associated with the findings, as long as the neural response latency is reasonable, because we found the best-fitted time delays for each sensory input and motor output from individual participants. Because we aimed to estimate the relative contributions of each retinal input and behavior-related component on the EEG activity pattern, we normalized $S_{IRVE}$ and $M_{vel}$ across different conditions, trials, and time points using max-min normalization. To avoid the estimation procedure falling into local minima, we iterated the fitting procedure 400 times by randomly shuffling the initial conditions. Among the 400 estimates, we selected the model that explained the variance of the data the most. The estimates of $w_s$, $w_m$ and $t_s$, $t_m$ were averaged across all the participants (Fig. 6b, c).

**Statistics and reproducibility**. Multiple comparison problems in the time-series statistical test were corrected using a nonparametric cluster-based permutation test[74]. Under the assumption that close time point samples are correlated, significant time clusters were selected from samples for which $p$ values of the consecutive $t$-values exceeded a certain cluster-defining threshold (e.g., $p<0.01$), and we computed the sum of $t$-values within each time cluster. Then, to test if the observed time clusters were significantly different from zero, the permutation distribution of summed $t$-values was computed by randomly shuffling the data with zeros (50,000 times) for each time cluster. Finally, the corrected $p$ values for the observed time clusters were determined by calculating the proportion of random partitions in the permutation distribution that resulted in a larger test statistic than the observed one.

**Reporting summary**. Further information on research design is available in the Nature Research Reporting Summary linked to this article.

## Data availability

Preprocessed EEG and eye velocity datasets can be accessed from the link, https://semoconlab.com/codes/reliability-weighted-information-integration/.

## Code availability

The analysis code is publicly available at https://www.dropbox.com/scl/fo/hnydr4ieldaqaxdom4asn/h?dl=0&rlkey=glkg1b5cj6libdvtvjlahv5om and archived to

Zenodo: https://doi.org/10.5281/zenodo.7529737. The code was written in Matlab (version 2019b).

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

## Acknowledgements
This study was supported by the Institute for Basic Science [IBS-R015-D1]. We thank Hyomin Yu and Kyungmin Kim for assisting with data collection. We also thank Dr. Yee-Joon Kim and Dr. Min-Suk Kang for constructive comments on earlier versions of this manuscript.

## Author contributions
Conceptualization, S.K. and J.L.; Methodology, W.J., S.K., and J.L.; Investigation, S.K. and J.P.; Formal analysis, W.J. and J.L.; Writing—original draft, W.J. and J.L.; Writing—review and editing, W.J., S.K., J.P., and J.L.; Funding acquisition, J.L.

## Competing interests
The authors declare no competing interests.
