## [Peer Review File · Communications Biology]

Reviewers' comments:

Reviewer #1 (Remarks to the Author):

Jeong et. al. examined how cueing direction of motion is integrated with target motion to drive smooth pursuit eye movements. Through an analysis of eye movements and EEG activity, they observed neural and behavioral evidence of Bayesian information integration. The study is well designed, the results are compelling and the conclusions are convincing. I only have some minor comments.

1. The results of the EEG seem somewhat unspecific. The authors find a nice behavioral effect; therefore, it is not surprising that somewhere in the brain there will be a neural signature of this difference. I understand that it is very hard to find these signatures with EEG but the conceptual merit was unclear. Presenting an alternative hypothesis on the expected EEG results or finding some specificity in the spatiotemporal properties of the EEG would further strengthen the paper.

2. Removal of eye movement artifacts. Some of the arguments concerning the artifact need to be clarified.

a. Removal of microsaccades indeed excludes the artifact they cause, but the EMG related to smooth pursuit eye movement might also lead to an artifact in the EEG. Has anyone quantified the smooth pursuit EMG artifact?

b. The argument about the latency is not clear. The EMG artifact has a short latency effect on the EEG (see for example, Greenberg, 2008 Figs. 2, 6, Dimigen 2009 Fig. 1) so that the EMG artifact at movement onset (~100 ms) is probably within the time of the analysis.

c. Could you please clarify the argument in the last paragraph of 4.3 "Moreover, if the EEG responses reflected eye movements ..."? During pursuit initiation, it seems that when the difference in eye movements between valid and non-valid conditions is maximal, it is also maximal in the EEG.

3. The same type of stimulus was used for cueing and driving eye movements. Is the behavioral effect related to the sensory properties of the cue? For example, could the cue lead to motion adaptation or priming? Alternately, the cue effect might not be linked to the specific sensory properties of the cue. In this case I would expect the effect of the cue to generalize to other types of cueing such as verbal instructions or other arbitrary cues. Please discuss or provide evidence.

4. The block design might mask the cue effect since the subjects might learn to ignore the cue in the invalid blocks. Did the authors consider probing this by using a rare invalid cue in a valid block? Perhaps inspecting the first trials in invalid blocks after a valid block might reveal a larger cue effect.

5. EEG linear model: Strong correlations between regressors might lead to overfitting. What is the correlation between the regressors? Specifically, are velocity error and eye velocity correlated (or negatively correlated)? Is eye position correlated with IRVE? Adding time lags might further increase the correlation between regressors; thus, what is the correlation of the regressor in the model after the time adjustment?

6. In Figure 1F, the time from movement latency is used to evaluate the effect of the cue. But the difficulty in the estimation of movement latency could confound the results. One hint of this problem is that at movement onset the red trace in 1F is already different than zero.

7. There is no subject-by-subject analysis. It would strengthen the paper to include at least some plots of the subject-by-subject analysis.

8. The authors conclude that: "sensory input drove the EEG activity pattern, and this EEG activity pattern drove motor behavior." This causal statement seems too strong. What is the evidence that the EEG activity that was measured indeed drove the motor behavior?

Reviewer #2 (Remarks to the Author):

In this study, using a smooth pursuit eye movement task, the authors examined if the process of reliability-based optimal information integration occurred in sensorimotor behavior and concurrent neural activity. They concluded that their results demonstrated that the effect of motion cues on the neural and behavioral responses to pursuit targets is more evident when the target motion is weak

and unreliable, which were consistent with previous studies and supported neural and behavioral evidences of Bayesian information integration. Their approach is interesting but there are some concerns in the methodological parts.

1. The detailed explanations of statistical analysis are missing. It seems that the authors did not perform multiple comparison corrections, so the reviewer cannot judge if the all results in the manuscript are reliable or not. there are a few parts in the figure legends saying 'corrected', but they did not explain how they perform it if they actually did it. It is also necessary to explain why the two statistical tests, t-test and permutation test, were used differently for each comparison.

2. Relating to the previous comment, the authors mentioned that Fig. 2A and 2B showed significant difference of onset between the blue and red signals, and solid and dotted lines, but looking at the figures, the difference looks is quite small and it does not look like there are statistical significance. In addition, there are no explanations for green and yellow line in Fig. 2A and 2B in the main text.

3. In preprocessing of data in Section 2.3, the author mentioned that saccadic eye movements were visually screened and discarded. How can the authors justify that this process is not arbitrary since they have not set up an algorithm for the discard? In Figure 1B, it seems that they wanted to explain that saccadic eye movements were discarded, but they just show example traces after removing the saccadic eye movements. If they would like to show the discard in the figure, it would be better to show traces before the discard.

4. There are several ambiguous expressions scattered in the throughout manuscript. For example, recalibration of eye position in 2.2 Data acquisition, the criteria for the recalibration were not numerically defined but just 'considerably away from the fixation point'. This is another concern about arbitrariness. Another ambiguous expression is regarding the brightness of the room. Instead of the expression 'reasonably dark room', a value for the brightness of the room should be added.

5. The explanations of the experimental paradigm are not enough, too. Please explain how the authors instructed the paradigm to the participants. Were the participants informed that there are valid and invalid conditions, and existence of the invisible square window? What is the purpose of the square? What is the aperture here? They often changed expressions to mention identical thing (e.g., the pursuit target and random dot patch and other things, too), so that causes misleading and hard to follow.

6. Relating to one previous comment, please define what is the open-loop and closed-loop periods in the manuscript. The explanation is missing. In addition, were the black arrows in Fig. 1A actually presented to the participants? Although the arrow is depicted in the figure, looking at the text, it seems as if it was not actually displayed.

7. Were the values of the Mahalanobis distance calculated using the EEG voltage values at individual time points? The authors used the term 'activities', but it does not adequately explain it. They seemed to use smoothed EEG signals by a 20ms rectangular time window to calculate the distance, so it would be better to show the smoothed signals for the individual condition.

8. On page 10, to calculate the Mahalanobis distance using the EEG signals, they repeated the iteration process for 10 times and calculating the average distance. Given the large variability in the EEG signal, the 10 repetitions might be insufficient. Therefore, to demonstrate the validity of the results in the current manuscript, it should be shown that the Mahalanobis distances converge without significant difference at more than 10 iterations.

9. Calculating the Mahalanobis distance for each channel location or brain region could have yielded more fruitful results. The authors mentioned that existing studies have revealed contributions of the sensorimotor cortices and the middle temporal areas, but why they did not calculate the Mahalanobis

distance values for individual EEG channels, even if the spatial resolution of EEG is low?

10. On page 11, they used the 300 ms duration of EEG data for the model estimation, from 31 to 330 ms, relative to stimulus motion onset. Please provide a reason to justify, rather than arbitrarily, the choice of this time frame.

11. On page 13, they explained that Fig. 1C and 1D were created using data between 0 and 100ms, but in the figure legend, it is explained that they used data between -100 and 100 ms. There are such inconsistencies or unclear expressions in other parts, too. Please check the whole manuscript carefully.

12. On page 13, subsequent to Section 3.2, why did the author only performed comparisons between the two angle only (0° and -60°) even though they performed the experiments using the three conditions? The results obtained from the analysis should be also shown even including the condition of -30° if the results are significant enough, which will make the manuscript more reliable. The same comment applies to the results in Section 3.3.

13. On page 18, in the linear model, they used eye velocity as the motor behavior and IRVE as the sensory input because they thought they were similar in shape to the EEG dissimilarity. However, this rationalization is not enough to investigate their purpose. If the other parameters cannot explain the EEG pattern dissimilarity well, such results might be evidence to support their hypothesis.

Minor comments

1. The words 'subjects' and 'participants' are mixed up and should be unified into 'participants' as they are human participants.

2. The figures are not arranged in the order they appear in the text and should be adjusted. Readers will need to go back and forth as they read. Figure 1B appears first in the current manuscript.

3. Definitions of the terminology do not seem to be arranged in the order they appear in the text. For example, the integrated retinal velocity error is not accounted for at the time of the initial description and seems to have been explained later.

4. I think Section 2.4. 'Stimuli and task design' should come before Section 2,2 'Data acquisition' to make readers understand this study easier. Also, it would be better that some explanations shown in Results section move to 'Stimuli and task design' section.

5. The figures are not overall friendly explained. For example, captions for red and blue lines and solid and dotted lines are not put in each figure, and the explanations are not enough to help readers understand. Fig. 1C and D also lack the legends.

Reviewer #3 (Remarks to the Author):

This manuscript reports an original and interesting study on reliability-based combination of retinal (low and high contrast target motion) and extraretinal (valid/invalid motion cue preceding the target) information. The effects of this combined information are measured on smooth pursuit eye movements and on the EEG multivariate recordings in human volunteers. The manuscript is overall clear and well written, although it would deserve a further reading to clean up some unclear parts. Results are nicely illustrated and statistical analyses seem appropriate. By analyzing the multivariate EEG disparity pattern (Mahalanobis distance across the multivariate multichannel recordings) across conditions, the authors conclude that there is evidence in favor of reliability based integration of information, whereby the validity of the motion cue would influence visual motion tracking but only (or much more prominently) when the visibility of the moving target is low. However, the EEG dissimilarity measure turns out to be most powerfully explained by the time-integrated retinal velocity speed, a fact that could strongly hide other more subtle effects of predictive information, leaving the reader in a sort of "not completely convinced" state. I outline some issues below.

Major issues

Participants exclusion: "Data from two participants were excluded because of insufficient trials (data from only two days were available)": was this really necessary? Did the authors try to include data from these two participants (using the appropriate statistical analysis)?

Test of the Bayesian integration hypothesis on eye movements: the effect of each invalid cue on smooth pursuit tracking should be presented specifically for each cue and target direction condition, not only as average across the two possible invalid cue directions (e.g. as in Figure 1C-D-E). This illustration (and related analyses) would provide a stronger quantitative support to the hypothesis of Bayesian integration of sensory and predictive information. Likewise, the condition-specific analysis of EEG dissimilarity (in terms of Mahalanobis distance) could further strengthen the support for this hypothesis. Just to make one example, would the deviation measured in the invalid block be larger (say for the target direction = 0deg) if the previous cue was moving at -60deg compared to -30deg? But also: would the oculomotor and EEG measured activities vary between the trials with identical target direction (and contrast) which follow a cue moving in the same direction, either during the valid cueing block or (in one third of trials) in the invalid cueing block?

Analysis of EEG pattern dissimilarity: the presentation of the EEG data is rich and interesting, but not always straightforward to follow. Each condition comparison (different target motion comparison, same target direction but different cue direction etc...) should be more clearly presented and the rationale better explained. In addition, some additional specific information about the multidimensional data analysis should be provided. For instance, why didn't the authors try to perform some form of source localization, and/or ICA/PCA dimensionality reduction, rather than going for the Mahalanobis distance (which should be presented in more detail, also) across all channels? A multivariate analysis seems a good idea but a reduction of dimensionality would intuitively improve the quality and specificity of results, am I correct? Did the authors analyse the residuals after the implementation of the model illustrated in Figure 5.A? After all, the information carried by the EEG signals about the cue-related account of reliability-based information is limited to the very initial part of the recordings. In this sense it would be worth providing all available evidence in favour of the Bayesian integration hypothesis to be more convincing.

Line 387: The rationale for the assumption that the integrated retinal velocity error is a more meaningful quantity than the retinal velocity error, for oculomotor initiation is not clear. What model of visuomotor transformation are the authors assuming? Classically, smooth pursuit acceleration is modelled as the result of a negative feedback loop minimizing the retinal velocity error... In the discussion the authors argue that the integration of the retinal velocity error is needed to estimate the target position error at the end of the open loop (to prepare the catch up saccade). Again, this appears as a sort of post-hoc explanation and the reader would need a clearer presentation of the initial hypotheses to be more convinced.

By the way, given the proposed conclusion that integrated rather than instantaneous retinal velocity error informs the informative multivariate EEG data, shouldn't the title be modified accordingly?

Minor points

Abstract : the sentence "we independently manipulated the strength of sensory motion in the pursuit target and direction of motion cue that was informative or uninformative for the subsequent pursuit target direction" is unclear and needs multiple readings to understand , please rephrase.

Likewise, the expression, "the information content that constitutes multivariate EEG direction discrimination..." is weird, maybe it should be changed into something like "the information content related to direction discrimination in multivariate EEG ..."

Introduction, line 35: "more efficient and optimal" seems redundant

Caption of Figure 1: there is a typo in the sentence (and elsewhere) "Each triangle indicates the average timing of the eyes to begin tracing the stimulus (latency)" Overall the authors should improve the clarity and accuracy of this Figure caption (e.g. "shaded areas" are sometimes mentioned while I don't see one...and other things).

Methods: the three possible direction of the moving target are sometimes described as (0° , -30° , and -60°) and sometimes as (0° , 300° , and 330°): please use a uniform notation and possibly add a schematic illustration of the three possible directions in the first figure

Results section: the initial part of this section contains information that should rather be (and in part it already is) in the Methods section. Please reorganize.

P. 16, lines 340-342: this sentence is ill-formulated "Reliability-weighted information integration would occur if the effect of the cue direction on the behavioral and EEG responses was greater when the pursuit target contrast was low" and should better be rewritten "One of the predictions of reliability-weighted information integration is that the effect of the cue direction on the behavioral and EEG responses is greater when the pursuit target contrast is low"

Reviewers' comments:

Reviewer #1 (Remarks to the Author):

Jeong et. al. examined how cueing direction of motion is integrated with target motion to drive smooth pursuit eye movements. Through an analysis of eye movements and EEG activity, they observed neural and behavioral evidence of Bayesian information integration. The study is well designed, the results are compelling and the conclusions are convincing. I only have some minor comments.

1. The results of the EEG seem somewhat unspecific. The authors find a nice behavioral effect; therefore, it is not surprising that somewhere in the brain there will be a neural signature of this difference. I understand that it is very hard to find these signatures with EEG but the conceptual merit was unclear. Presenting an alternative hypothesis on the expected EEG results or finding some specificity in the spatiotemporal properties of the EEG would further strengthen the paper.

Thank you for providing us with an opportunity to strengthen our paper. Following the reviewer's suggestion, we sought to investigate if the Bayesian integration of cue and pursuit is limited to a specific region in the brain. We divided electrodes by the locations along the anterior-posterior axis, then we performed the same analysis on each subgroup. We found that the effect of Bayesian inference mostly happened in vertex channels. This result suggested that the prior expectation of motion direction might be integrated with the sensory input in the parietal areas. This result provided a clue about where the integration of prior expectation with sensory input is likely to occur in the brain. In the manuscript, we have included this result in Figure 4 (Supplementary Figure 3): lines 254–269, 418 - 444.

Fig. 4

Supplementary Fig. 3

2. Removal of eye movement artifacts. Some of the arguments concerning the artifact need to be clarified.

a. Removal of microsaccades indeed excludes the artifact they cause, but the EMG related to smooth pursuit eye movement might also lead to an artifact in the EEG. Has anyone quantified the smooth pursuit EMG artifact?

The typical spatial resolution of EOG (EMG) is worse than 1 deg, which means that discrimination of eye position or velocity using muscle activity would be possible only when the displacement of eye position is huge. In our study, almost all the EEG result was obtained by comparing EEG activity across different motion direction or cue conditions. And, our analysis focused on the initiation of pursuit where the effect of EOG (EMG) on the EEG activity will be pretty small; at 15 deg/s target speed, eyes would be displaced less than 1.5 deg at 200 ms from the motion onset. Our EEG evidence of the Bayesian information integration appeared earlier than ~150 ms, where the eye displacement would be less than 1 deg. The eye position difference between the two direction conditions would be even less than that. We also included an ICA-based eye movement-related artifact removal in

the preprocess pipeline (please see the methods). This routine is not perfect, but the apparent artifacts that were induced by eye muscle activity should have been taken care of.

b. The argument about the latency is not clear. The EMG artifact has a short latency effect on the EEG (see for example, Greenberg, 2008 Figs. 2, 6, Dimigen 2009 Fig. 1) so that the EMG artifact at movement onset (~100 ms) is probably within the time of the analysis.

It is true that the effect of EMG artifacts on EEG activity is immediate. However, we believe this artifact did not influence the conclusion of our study. First, the EEG modulation by prior expectation only occurred when the stimulus contrast was low. If the EEG modulation was mainly induced by the EMG artifact, a similar or stronger effect should be observed when stimulus contrast was high since the initiation of smooth pursuit by the high contrast stimulus would be stronger and faster. Second, the EEG modulation happened in advance of the eye movement initiation. When stimulus contrast was high, the average pursuit latency was ~90 ms, but when stimulus contrast was low, the pursuit latency was > 120 ms (see Figure a. below). Therefore, the effect of prior expectation that we reported happened before motor movement initiation.

Figure a. Average pursuit latency in cue valid (A) and cue invalid (B) conditions. Blue shows the average latency when the contrast of the pursuit target was 100%, red shows the average latency when the contrast was 12%.

c. Could you please clarify the argument in the last paragraph of 4.3 “Moreover, if the EEG responses reflected eye movements...”? During pursuit initiation, it seems that when the difference in eye movements between valid and non-valid conditions is maximal, it is also maximal in the EEG.

Sorry for the confusion. The comparison here was done based on different directions of motion instead of the difference in cue validity. Eye velocity difference (Mahalanobis distance) between the two directions (0 deg vs. -60 deg) is smaller in the cue-invalid condition than in the cue-valid condition. However, Mahalanobis distance of EEG activity patterns in the same two direction conditions was larger in the cue-invalid condition than in the cue-valid condition (please see Figure 2A). We corrected the manuscript to prevent confusion (lines 513- 521).

3. The same type of stimulus was used for cueing and driving eye movements. Is the behavioral effect related to the sensory properties of the cue? For example, could the cue lead to motion adaptation or priming? Alternately, the cue effect might not be linked to the specific sensory properties of the cue. In this case I would expect the effect of the cue to generalize to other types of cueing such as verbal instructions or other arbitrary cues. Please

discuss or provide evidence.

Both the validity and motion direction of the cue affects the following pursuit eye movements. In the cue-invalid condition, the direction of the cue matches the pursuit target direction in 1/3 of the conditions. When we compared the pursuit traces in the cue-valid condition and the cue-invalid condition only in the cases the cue direction matches the pursuit target direction, pursuit directions were biased toward the central direction more in an invalid condition than in a valid one (Supplementary Figure 1). Therefore, there is an effect of cue validity regardless of cue directions. And, as has been explained in the paper, the pursuit directions were attracted toward the cue direction when the cue direction did not match the target direction. So, two types of prior expectations are working in our experiment. One is prior knowledge learned from the validity of the cue, and the other is prior knowledge implicitly formed from the sensory stimulus. The effect of the cue validity was evident in behavioral results, but the effect on EEG representations was noisy and indistinguishable, probably because of the small sample size when trials in the cue-invalid condition were divided.

Supplementary Fig. 1

4. The block design might mask the cue effect since the subjects might learn to ignore the cue in the invalid blocks. Did the authors consider probing this by using a rare invalid cue in a valid block? Perhaps inspecting the first trials in invalid blocks after a valid block might reveal a larger cue effect.

Thank you for pointing out a very interesting aspect of the study. Unfortunately, we did not have a rare invalid cue condition. Dividing trials would be a good alternative, but it was also hard to identify the effect in EEG analysis due to the small number of trials. In the multivariate analysis that we used, we need quite a lot of trials for getting robust results. Therefore, it was difficult for us to obtain a robust EEG representation of the sensory and cognitive factors when dividing trials. We were able to analyze and compare the pursuit behaviors in the first and the last a few trials though. We selected the first 20 trials and the last 20 trials in each block and averaged the eye velocity traces. In this behavioral analysis, we could not find a clear trend or evidence showing the effect of the previous block on the early trials in the next block (please see Figure b below).

Figure b. Eye velocity traces in the first and the last 20 trials in each block.

5. EEG linear model: Strong correlations between regressors might lead to overfitting. What is the correlation between the regressors? Specifically, are velocity error and eye velocity correlated (or negatively correlated)? Is eye position correlated with IRVE? Adding time lags might further increase the correlation between regressors; thus, what is the correlation of the regressor in the model after the time adjustment?

We understand the reviewer's concern regarding the model. Because velocity errors were obtained from the difference between retinal image motion and eye velocity, naturally eye velocities were correlated with retinal velocity errors if we look at the temporal profiles of each regressor. Also, the position itself will be correlated with the IRVE because the temporal evolution would be similar to each other. In our model fitting, time lags were included as parameters to be estimated, therefore, the regressors themselves were determined by the best-fitted (model that explained the EEG pattern the most) time lags. As a result, we only had the time-adjusted regressors and they were positively or negatively correlated with each other. Partly because of this innate correlation

between IRVE and position, eye velocity and velocity error, we used the reduced model instead of the full model in the paper. Indeed adding eye position or velocity error did not improve the explainability of the model, which might be because of the innate correlation.

An important test here to know if the innate correlation caused any problem in the model estimation, was whether the weights assigned to the individual templates were correlated with each other across different participants. We found no correlation between the weights, suggesting that they were playing different roles in explaining individual EEG data (Supplementary Figure 5). Now, we have explained the points that the reviewer raised up in the paper to help the reader to understand the model better (lines 332 - 337).

Supplementary Fig. 5

6. In Figure 1F, the time from movement latency is used to evaluate the effect of the cue. But the difficulty in the estimation of movement latency could confound the results. One hint of this problem is that at movement onset the red trace in 1F is already different than zero.

Thank you for the comment. A critical issue, in this case, would be if the significant difference appeared because pursuit latency estimation in the low contrast condition was more conservative (estimated latency was larger than the true latency). Then, the effect would be higher in the low contrast condition only because we compared the effect in a different time window across contrast conditions. To address this potential problem, we set the latency in low contrast 20 ms earlier than the estimated latency. Even in this condition (see Figure c. below), the effect in low contrast was higher than the effect in high contrast. Because it is highly unlikely for us to set the behavioral latency 20 ms later than the true one, we argue that our conclusion would not be altered by the error in behavioral latency estimation in the low contrast condition.

Figure c. Differences in the eye velocity trace distance between valid and invalid blocks ($y_2 - y_1$) between -100 and 100 ms from pursuit latency. Here, we set the latency 20 ms faster than the originally estimated latency. The bar graphs on the top left show the distance differences at 100 ms after latency within the open-loop period of the smooth pursuit. The shaded areas and error bars show the standard errors. $*p < 0.05$.

7. There is no subject-by-subject analysis. It would strengthen the paper to include at least some plots of the subject-by-subject analysis.

We sought to find any subject-by-subject effects and correlations, but it was hard for us to find them. The subject-by-subject analysis is possible when there is quite a good amount of variability across the participants in both behavioral effects and multivariate EEG representations, and when the signal-to-noise ratio of the neural data is high enough. In this experiment, getting a robust EEG representation of the Bayesian information integration was hard given the number of conditions used in the experiment and the number of trials. We had to ask participants to visit us three times to get the current level of signal-to-noise ratio in the multivariate EEG representation. We might need much more data to achieve the required quality of the EEG representation for the individual analyses, which was practically hard to do. We hope the reviewer understands the properties of our data and situation.

8. The authors conclude that: "sensory input drove the EEG activity pattern, and this EEG activity pattern drove motor behavior." This causal statement seems too strong. What is the evidence that the EEG activity that was measured indeed drove the motor behavior?

We agree with the reviewer. We replaced the expression with the weaker one (lines 327 – 328).

Reviewer #2 (Remarks to the Author):

In this study, using a smooth pursuit eye movement task, the authors examined if the process of reliability-based optimal information integration occurred in sensorimotor behavior and concurrent neural activity. They concluded that their results demonstrated that the effect of motion cues on the neural and behavioral responses to pursuit targets is more evident when the target motion is weak and unreliable, which were consistent with previous studies and supported neural and behavioral evidences of Bayesian information integration. Their approach is interesting but there are some concerns in the methodological parts.

1. The detailed explanations of statistical analysis are missing. It seems that the authors did not perform multiple comparison corrections, so the reviewer cannot judge if the all results in the manuscript are reliable or not. there are a few parts in the figure legends saying 'corrected', but they did not explain how they perform it if they actually did it. It is also necessary to explain why the two statistical tests, t-test and permutation test, were used differently for each comparison.

We apologize for the confusion. We have performed all the statistical analyses with multiple comparison correction whenever necessary. When we did not need to correct the multiple comparison, we used t-test. However, we agree with the reviewer that a detailed explanation for the statistical test was missing in the current manuscript. In the method section, we explained the statistical method that we used in this study (lines 682 – 690).

2. Relating to the previous comment, the authors mentioned that Fig. 2A and 2B showed significant difference of onset between the blue and red signals, and solid and dotted lines, but looking at the figures, the difference looks is quite small and it does not look like there are statistical significance. In addition, there are no explanations for green and yellow line in Fig. 2A and 2B in the main text.

We again apologize for the confusion. In those figures, we sought to test if the Mahalanobis distance in each contrast condition was significantly different from zero, using a cluster-based permutation test (multiple comparison correction was made here). If the Mahalanobis distance is significantly different from zero, it means that the neural direction discrimination between the two motion stimuli would be significant. We here reported that the significant neural direction discrimination was faster in the valid condition, just based on the timing when the significant neural direction discrimination started. A direct comparison between valid and invalid conditions was done in Figure 2C. We modified the manuscript to prevent further confusion. We also included an explanation for the behavioral data (green and yellow lines) in the result (lines 186 – 193).

3. In preprocessing of data in Section 2.3, the author mentioned that saccadic eye movements were visually screened and discarded. How can the authors justify that this process is not arbitrary since they have not set up an algorithm for the discard? In Figure 1B, it seems that they wanted to explain that saccadic eye movements were discarded, but they just show example traces after removing the saccadic eye movements. If they would like to show the discard in the figure, it would be better to show traces before the discard.

We applied the semi-automation for removing trials with saccadic eye movements. During fixation, if the spontaneous eye speeds in a time window between -100 ms and 50 ms relative to motion onset exceed 5 deg/s, we automatically removed the trials. After eyes started moving, this absolute criterion was not suitable because the pursuit latency, speed, etc are different across participants. We could have used a combination of velocity and acceleration criteria for detecting big saccades, but the application of these criteria often missed subtle saccades. If we used the objective criteria, we usually did the manual screening again to remove trials with smaller saccades that were not detected by the objective criteria. Therefore, we did the screening with the eyes from the start. We

hope the reviewer understands the special situation in the research of smooth pursuit eye movements. We included a figure that showed the eye traces before saccade removal below.

4. There are several ambiguous expressions scattered in the throughout manuscript. For example, recalibration of eye position in 2.2 Data acquisition, the criteria for the recalibration were not numerically defined but just 'considerably away from the fixation point'. This is another concern about arbitrariness. Another ambiguous expression is regarding the brightness of the room. Instead of the expression 'reasonably dark room', a value for the brightness of the room should be added.

In this experiment, we have higher eye position monitoring standards than in other EEG studies. During the fixation duration, if eye positions deviated from 2 degrees square window (invisible), the trial was aborted and switched to the next trial. Therefore, participants worked hard to maintain their fixation. However, when the participant changed his/her head position because they could not maintain it after the long-term experiment, eyes became constantly off from the 2 degrees square window, and trials got aborted repeatedly. Then, we recalibrated the eye tracker to compensate for the changes in head position. The recalibration is just to adjust the absolute offset of the eye position; therefore, it did not affect the quality of the data or the conclusion of the paper (they were mainly from relative differences or temporal changes). It is true that whether to do the recalibration procedure or not is the experimenter's arbitrary decision, but it is also not practical to define objective criteria for the recalibration and implement this in the experiment control program. The intervention of the experimenter is usually more efficient and productive in making the experiment keep going. We hope the reviewer understands the purpose of the recalibration that we performed, and the reason for not using the objective, automated criteria. We added a brief explanation to the method section to prevent confusion (lines 600 - 603).

Sorry for the confusion regarding the brightness of the room. We said 'reasonably dark' because there were always light from the monitor and the monitor was the only light source. Because we reported the luminance

range of the monitor, and the participant's vision would be most affected by the level of light that we reported (we measured the brightness of the monitor from the point where the participant's head would be located), we did not report the general light level in the room (the measured levels of the light would depend on locations in the room). We revised the manuscript to prevent confusion (lines 566 – 569).

5. The explanations of the experimental paradigm are not enough, too. Please explain how the authors instructed the paradigm to the participants. Were the participants informed that there are valid and invalid conditions, and existence of the invisible square window? What is the purpose of the square? What is the aperture here? They often changed expressions to mention identical thing (e.g., the pursuit target and random dot patch and other things, too), so that causes misleading and hard to follow.

We revised the manuscript so that consistency in the usage of terms is maintained. Also, included a more detailed explanation of the experimental design, verbal instruction given to the participants, etc. The invisible square window was used to guarantee a good fixation and pursuit. If the participant's eye deviated from the window, we aborted trials because the participant did not fixate at the fixation point properly or did not track the pursuit target well. Participants were verbally informed of the existence of the invisible fixation window. We included these explanations in the manuscript (lines 557 – 593, and the results).

6. Relating to one previous comment, please define what is the open-loop and closed-loop periods in the manuscript. The explanation is missing. In addition, were the black arrows in Fig. 1A actually presented to the participants? Although the arrow is depicted in the figure, looking at the text, it seems as if it was not actually displayed.

We are sorry for omitting the explanation. We added a brief definition of open-loop and closed-loop in the manuscript (in the Result section). Usually, the open-loop period is ~100 ms after pursuit initiation, and the following period is termed as the close-loop period, which has been identified in previous studies. During the open-loop, the pursuit system did not get feedback about its relevant position to the visual target because it was too early for the feedback signal to reach the system (lines 123 – 130).

The arrows displayed in the figure were never displayed to the participants, but were used to explain the experiment better. The arrows only indicate the motion direction of the random dot patch. We have explained the illustration in the figure caption to help readers to understand the manuscript better (Fig.2, Fig. 3, Fig. 7 captions)

7. Were the values of the Mahalanobis distance calculated using the EEG voltage values at individual time points? The authors used the term 'activities,' but it does not adequately explain it. They seemed to use smoothed EEG signals by a 20ms rectangular time window to calculate the distance, so it would be better to show the smoothed signals for the individual condition.

We are sorry for the confusion that originated from the insufficient explanation. The Mahalanobis distance is a measure of activity pattern difference between conditions normalized by the covariance matrix, and we used the voltage values for the calculation. Because it is a standardized value that showed the difference in EEG activity patterns, we used the term 'activity.'

When we calculated the Mahalanobis distance, we first smoothed the EEG voltage with 20 ms rectangular time window (please see Methods, lines 643 - 648). The smoothed EEG was used in calculating the Mahalanobis distance, therefore, what we have shown in the manuscript are the plots of Mahalanobis distances calculated from smoothed EEG with 20 ms rectangular time window. They are indeed the smoothed signals.

8. On page 10, to calculate the Mahalanobis distance using the EEG signals, they repeated the iteration process for

10 times and calculating the average distance. Given the large variability in the EEG signal, the 10 repetitions might be insufficient. Therefore, to demonstrate the validity of the results in the current manuscript, it should be shown that the Mahalanobis distances converge without significant difference at more than 10 iterations.

This iteration was performed to match the number of trials used in each condition when calculating Mahalanobis distance since the Mahalanobis distance calculation can have a bias when the number of trials between the conditions is largely different. In our data, the trial difference between conditions was small, therefore, the bias due to the trial number difference was not likely to happen. However, we did an iteration procedure to remove any possibility of the bias originating from the small trial number differences. For this purpose, we think 10 iterations is enough and the analysis using 1000 iterations gave almost identical results. We showed the results with 1000 iterations as in the Figure below (identical analysis with Figure 2, 1000 iterations).

9. Calculating the Mahalanobis distance for each channel location or brain region could have yielded more fruitful results. The authors mentioned that existing studies have revealed contributions of the sensorimotor cortices and the middle temporal areas, but why they did not calculate the Mahalanobis distance values for individual EEG channels, even if the spatial resolution of EEG is low?

Thank you for the suggestion. Following the reviewer's suggestion, now we calculated the Mahalanobis distance of subgroups of channels along the anterior-posterior axis. With this subgroup analysis, we indeed yielded more fruitful results (please see our response to reviewer 1's question). We did not perform the Mahalanobis distance calculation on each channel because the Mahalanobis distance calculation is one of the multivariate analysis methods that were developed to overcome the limitations of the univariate approach. By analyzing EEG activity patterns across multiple channels, we were able to obtain more sensitive measures.

10. On page 11, they used the 300 ms duration of EEG data for the model estimation, from 31 to 330 ms, relative to stimulus motion onset. Please provide a reason to justify, rather than arbitrarily, the choice of this time frame.

The purpose of this modeling was to know if retinal input and behavioral output substantially contributed to the temporal change pattern of EEG activity. Therefore, we selected a time window that can capture the dynamic changes in the EEG activity pattern. Here the selection criterion was to capture the open-loop smooth pursuit eye movement (the nominal time window for the open-loop smooth pursuit is between 100 ms and 200 ms from target motion onset). So, the first 100 ms from motion onset would be the duration where retinal input would dominate, and the duration between 200 and 300 ms would be the duration where motor output would dominate because retinal input would be minimal (at the end of the open-loop, the eyes would closely track the motion target, which effectively removed the retinal input). Therefore, the total duration of interest for the pursuit behavior would be from 0 to 300 ms relative to motion onset. In the EEG activity, we had to consider the neural latency; therefore, we added a 30 ms delay on the 300 ms time window, which results in a time window between 31 and 330 ms from motion onset. A 30 ms delay is a reasonable choice, but is not defined by a quantitative criterion. However, as long as this time window captures the overall temporal profile of the EEG activity pattern, it would not have a substantial effect on the conclusion of the paper. The actual delay between the regressors and EEG activity is estimated individually using the fitting procedure. Therefore, individual differences already have been taken care of. We have added these arguments to the manuscript (lines 668 – 676).

11. On page 13, they explained that Fig. 1C and 1D were created using data between 0 and 100ms, but in the figure legend, it is explained that they used data between -100 and 100 ms. There are such inconsistencies or unclear expressions in other parts, too. Please check the whole manuscript carefully.

We are sorry for the mistake and inconsistencies that appeared in the paper. We tried hard to find such errors and corrected them as possible as we can. We hope that the manuscript is now well prepared for the journal.

12. On page 13, subsequent to Section 3.2, why did the author only performed comparisons between the two angle only (0° and -60°) even though they performed the experiments using the three conditions? The results obtained from the analysis should be also shown even including the condition of -30° if the results are significant enough, which will make the manuscript more reliable. The same comment applies to the results in Section 3.3.

We used the two outer directions because they showed the biggest behavioral and neural effects. We have done the analysis using other combinations with the smaller angle difference, and the overall pattern of the result was similar to the original analysis result, with the smaller effect. We did not include the data from the smaller angle difference as the main figure because the overall message and conclusion would not be different. Instead, we have included the result in Supplementary Figure 2 for the reviewer and the readers.

Supplementary Fig. 2

13. On page 18, in the linear model, they used eye velocity as the motor behavior and IRVE as the sensory input because they thought they were similar in shape to the EEG dissimilarity. However, this rationalization is not enough to investigate their purpose. If the other parameters cannot explain the EEG pattern dissimilarity well, such results might be evidence to support their hypothesis.

A simple assumption that we made in building this model was that the EEG activity pattern would be explained by the sensory input that the smooth pursuit system received and the motor output that the system would make. There would be other components that could explain the EEG activity pattern, for example, arousal level, attention, etc. However, we did not have objective measures for them that we can use for modeling the EEG data. Input and output of the system were obvious components that existed with objective measures (eye movements and visual motion). Therefore, we decided to use them for building this simple model. We initially constructed a full model with all possible inputs and outputs of the system (Supplementary Figure 4). However, this model did not substantially differ from the reduced model that we used in the manuscript where redundant retinal input and motor output components were omitted. Therefore, we suggested the reduced model as the main model in the manuscript. The way we explained the model might have induced the misunderstanding that our model building was just based on the shape of the possible model components. Therefore, in the revision, we explained our rationale for the components selection (lines 308 - 337). These modeling efforts let us identify which components of the input and output of the system dominated the EEG activity dissimilarity pattern. We also noted that the effect of the cue and the Bayesian integration of the sensory cue with the pursuit target, which happened before the eye movement initiation was not explained by the simple retinal input and motor output because the residual components that appeared after we subtracted the model prediction from the EEG pattern dissimilarity were largely intact (see Figure 7). Therefore, that component should have an extra-retinal origin that cannot be simply explained by the sensory input or motor output. We discussed these points in the manuscript (lines 338 – 370).

Fig. 7

Minor comments

1. The words 'subjects' and 'participants' are mixed up and should be unified into 'participants' as they are human participants.

We replaced 'subjects' with 'participants' throughout the manuscript.

2. The figures are not arranged in the order they appear in the text and should be adjusted. Readers will need to go back and forth as they read. Figure 1B appears first in the current manuscript.

We adjusted the Figure order whenever it is necessary. Hope it is easier to read now.

3. Definitions of the terminology do not seem to be arranged in the order they appear in the text. For example, the integrated retinal velocity error is not accounted for at the time of the initial description and seems to have been explained later.

Fixed them.

4. I think Section 2.4. 'Stimuli and task design' should come before Section 2,2 'Data acquisition' to make readers understand this study easier. Also, it would be better that some explanations shown in Results section move to 'Stimuli and task design' section.

We adjusted the order.

5. The figures are not overall friendly explained. For example, captions for red and blue lines and solid and dotted lines are not put in each figure, and the explanations are not enough to help readers understand. Fig. 1C and D also lack the legends.

In this case, we intentionally omitted the legend for making the figures simpler because the same legend exists in Fig. 1B and C. However, we agree with the reviewer that it might have been an oversimplification. We added the legends and the missing explanation in the captions.

Fig. 1

Reviewer #3 (Remarks to the Author):

This manuscript reports an original and interesting study on reliability-based combination of retinal (low and high contrast target motion) and extraretinal (valid/invalid motion cue preceding the target) information. The effects of this combined information are measured on smooth pursuit eye movements and on the EEG multivariate recordings in human volunteers. The manuscript is overall clear and well written, although it would deserve a further reading to clean up some unclear parts. Results are nicely illustrated and statistical analyses seem appropriate. By analyzing the multivariate EEG disparity pattern (Mahalanobis distance across the multivariate multichannel recordings) across conditions, the authors conclude that there is evidence in favor of reliability based integration of information, whereby the validity of the motion cue would influence visual motion tracking but only (or much more prominently) when the visibility of the moving target is low. However, the EEG dissimilarity measure turns out to be most powerfully explained by the time-integrated retinal velocity speed, a fact that could strongly hide other more subtle effects of predictive information, leaving the reader in a sort of "not completely convinced" state. I outline some issues below.

Major issues

Participants exclusion: "Data from two participants were excluded because of insufficient trials (data from only two days were available)": was this really necessary? Did the authors try to include data from these two participants (using the appropriate statistical analysis)?

When we included the data from the two participants, the overall pattern of the result was the same. However, some of the statistical tests became marginally significant. Also, the explainability of the linear model for these two participants was substantially poorer compared to other participants' model predictions. Therefore, we consider these two participants as outliers. We believe this is because of the quality of the Mahalanobis distance analysis done in EEG activity. Multivariate analysis, including the Mahalanobis distance analysis, requires lots of trials to get a reliable result. Because of this, we combined three days' data. Just having two days' data appeared to be insufficient for obtaining reliable neural signatures. We hope the reviewer understands our rationale for the decision.

Test of the Bayesian integration hypothesis on eye movements: the effect of each invalid cue on smooth pursuit tracking should be presented specifically for each cue and target direction condition, not only as average across the two possible invalid cue directions (e.g. as in Figure 1C-D-E). This illustration (and related analyses) would provide a stronger quantitative support to the hypothesis of Bayesian integration of sensory and predictive information. Likewise, the condition-specific analysis of EEG dissimilarity (in terms of Mahalanobis distance) could further strengthen the support for this hypothesis. Just to make one example, would the deviation measured in the invalid block be larger (say for the target direction = 0deg) if the previous cue was moving at -60deg compared to -30deg? But also: would the oculomotor and EEG measured activities vary between the trials with identical target direction (and contrast) which follow a cue moving in the same direction, either during the valid cueing block or (in one third of trials) in the invalid cueing block?

Thank you very much for the constructive suggestions. They are important and critical questions, revealing how prior knowledge and expectation affect the sensory information processes. Following the reviewer's suggestion (also please see our response to reviewer 1's query), we divided the trials in an invalid cue condition according to the difference between cue direction and pursuit target direction. In the behavioral analysis, we observed the effect of both the validity of the cue and the direction difference between cue and the pursuit target (Supplementary Figure 1). Therefore, both prior knowledge of the validity of the cue and cue direction modulated

the pursuit behavior. Interestingly, the pursuit direction was attracted towards the cue direction, and the amount of the attraction somewhat depended on the size difference between the cue and pursuit directions. We also wanted to see if the multivariate EEG representation follows behavioral changes. Although we found some effects of cue-target disparity on the EEG dissimilarity, we could not find a systematic relationship between the two and the EEG dissimilarity, when divided, was noisier. The absence of a systematic relationship might be because of the insufficient number of trials. The multivariate pattern analysis of EEG activity requires a lot of trials to get reliable neural representation, and dividing trials in cue-invalid conditions reduced the number of trials in each condition to 1/3 of the whole trials. Therefore, our failure in observing the systematic relationship between the cue-target difference and the corresponding EEG dissimilarity might be because of insufficient trials for the analysis, which makes it difficult for us to reach solid conclusions regarding more specific interactions between cue and target directions. Hope our new analysis and explanation addressed the reviewer's question.

Analysis of EEG pattern dissimilarity: the presentation of the EEG data is rich and interesting, but not always straightforward to follow. Each condition comparison (different target motion comparison, same target direction but different cue direction etc...) should be more clearly presented and the rationale better explained. In addition, some additional specific information about the multidimensional data analysis should be provided. For instance, why didn't the authors try to perform some form of source localization, and/or ICA/PCA dimensionality reduction, rather than going for the Mahalanobis distance (which should be presented in more detail, also) across all channels? A multivariate analysis seems a good idea but a reduction of dimensionality would intuitively improve the quality and specificity of results, am I correct ?

We apologize for the insufficient explanation for the analysis method and rationale. We revised the manuscript so that the method and the rationale to be clearly explained. We also explained the Mahalanobis distance analysis in more detail to help readers to understand the benefits of this multidimensional analysis.

We did not use the source localization approach because we did not take MRI image from each participant. We could have used the standard anatomical brain image for the source localization, but that would be an approximate measure that could easily mislead the readers. Instead, we divided the electrodes by locations across anterior-posterior axis and performed the same analysis to give an idea which brain regions would be dominant in the effect that we reported. Now, the result of this analysis has been included in Figure 4 and Supplementary Figure 3 in the manuscript (lines 254– 269, 418 - 444, and please see our response to reviewer 1's query).

Dimensionality reduction would be a good candidate for analyzing the multivariate pattern of the EEG activity. In this paper, however, we focused more on neural direction discrimination in different stimulus contrast and cue conditions. Therefore, we should compare the neural patterns across direction conditions regardless of whether we performed the dimensionality reduction or not. Here, we can use the Mahalanobis distance analysis on the neural data whose dimension has been reduced. Now the question is whether we compared the patterns for different stimulus conditions using all channels or reduced numbers of dimensions (PCA or ICA). We think the message would be the same eventually. We tried to explain the method and rationale for selecting this analysis in more detail (lines 174 – 179).

Did the authors analyse the residuals after the implementation of the model illustrated in Figure 5.A?

After all, the information carried by the EEG signals about the cue-related account of reliability-based information is limited to the very initial part of the recordings. In this sense it would be worth providing all available evidence in favour of the Bayesian integration hypothesis to be more convincing.

Thank you for an insightful suggestion. Following the reviewer's suggestion, we performed the residual analysis of the data (Figure 7). Using this analysis, we were able to bridge the modeling portion of the paper with the

Bayesian inference portion, which indeed made the paper more coherent. When we removed the sensory input and motor output components from the EEG dissimilarity pattern using the modeling approach, the remaining EEG activity pattern showed the components that were not accounted for by the simple input and output of the system. In this residual analysis, we found the more robust effect of Bayesian inference when pursuit target contrast was weak. This suggests that the EEG representation of the Bayesian integration was not the result of simple sensory input or motor output. Prior expectation of the motion modulated the sensory input, in a way that cannot be explained by the simple input to and output of the system. We discussed the meaning of this new result and emphasized that Bayesian neural modulation had an extraretinal origin (lines 338 – 370).

Line 387: The rationale for the assumption that the integrated retinal velocity error is a more meaningful quantity than the retinal velocity error, for oculomotor initiation is not clear. What model of visuomotor transformation are the authors assuming? Classically, smooth pursuit acceleration is modelled as the result of a negative feedback loop minimizing the retinal velocity error... In the discussion the authors argue that the integration of the retinal velocity error is needed to estimate the target position error at the end of the open loop (to prepare the catch up saccade). Again, this appears as a sort of post-hoc explanation and the reader would need a clearer presentation of the initial hypotheses to be more convinced.

Thank you for your valuable feedback. As the reviewer pointed out, the classical models of smooth pursuit assume retinal velocity errors as an input and velocity signal as an output (Krauzlis and Lisberger, 1994, Robinson et al., 1986, van den Berg, 1988). Therefore, we initially sought to model the EEG pattern dissimilarity with retinal velocity errors as inputs and eye velocity as an output. However, previous models originated from circuit analysis, which did not care much about the brain. In the brain, the eye movement itself is controlled by position signal generated from brain stem neurons (abducence nucleus, Nakamagoe et al., 2000), therefore retinal velocity errors should be transformed into motor command that controls the eye position. Moreover, we do not know what information would be represented in the EEG activity pattern. Therefore, we included possible inputs that could drive the EEG activity and motor outputs that could be explained by or controlled the EEG activity in the model. We found that the integrated retinal velocity input was the component that explained the EEG activity pattern the most, but it is difficult to know why this component was dominant. One possibility would be the integration of the retinal velocity error for the catch-up saccade, which has been already suggested to be included in the smooth pursuit model when the interaction between the smooth pursuit and catch-up saccade happened (Blohm et al., 2006, Nachmani et al., 2020). Accumulation of sensory evidence for the final decision-making has been shown previously in many studies, especially when saccadic eye movements were used as motor behaviors reporting the decision (Huk and Shadlen, 2005). The integration of the retinal velocity error was similar to this scheme; therefore, it is possible that the EEG activity pattern indeed represents this accumulation of sensory evidence for the saccadic eye movements (even if it is only speculative).

We included these arguments in the results and discussion of the revised manuscript so that the readers could have a clearer view and understanding of our model. We hope our rationale for the model building and regressor selection will be satisfactory to the reviewer (and to the readers): lines 308 – 337, 458 - 485.

By the way, given the proposed conclusion that integrated rather than instantaneous retinal velocity error informs the informative multivariate EEG data, shouldn't the title be modified accordingly?

Thank you for the suggestion. We included 'integrated' in the title of the paper.

Minor points

Abstract : the sentence "we independently manipulated the strength of sensory motion in the pursuit target and direction of motion cue that was informative or uninformative for the subsequent pursuit target direction" is unclear and needs multiple readings to understand , please rephrase.

We rephrased the sentence to make it easier to understand.

Likewise, the expression, "the information content that constitutes multivariate EEG direction discrimination..." is weird, maybe it should be changed into something like "the information content related to direction discrimination in multivariate EEG ..."

Thank you for the suggestion. We deleted the sentence to make the abstract more concise.

Introduction, line 35: "more efficient and optimal" seems redundant

We removed the 'optimal' from the sentence.

Caption of Figure 1: there is a typo in the sentence (and elsewhere) "Each triangle indicates the average timing of the eyes to begin tracing the stimulus (latency)" Overall the authors should improve the clarity and accuracy of this Figure caption (e.g. "shaded areas" are sometimes mentioned while I don't see one...and other things).

We are sorry for the tedious job in figure caption and thank you for pointing out these problems. We improved the clarity of the captions and corrected errors.

Methods: the three possible direction of the moving target are sometimes described as (0° , -30° , and -60°) and sometimes as (0° , 300° , and 330°): please use a uniform notation and possibly add a schematic illustration of the three possible directions in the first figure

We are sorry for the inconsistency. We used the same notation (0° , -30° , and -60°) and added a schematic illustration of the three possible directions in Figure 1.

Results section: the initial part of this section contains information that should rather be (and in part it already is) in the Methods section. Please reorganize.

Thank you for the suggestion.

Following the formatting guideline of the journal, we reorganized the manuscript in the revision. Now, the result section comes earlier than the method section, therefore, we explained essential aspects of the experimental design in the result so that readers can understand it better. We hope the reviewer understands our rationale for

doing this way, in light of the changed manuscript format.

P. 16, lines 340-342: this sentence is ill-formulated "Reliability-weighted information integration would occur if the effect of the cue direction on the behavioral and EEG responses was greater when the pursuit target contrast was low" and should better be rewritten "One of the predictions of reliability-weighted information integration is that the effect of the cue direction on the behavioral and EEG responses is greater when the pursuit target contrast is low"

Thank you for the correction. We corrected the sentence following the reviewer's suggestion (lines 227 – 229).

REVIEWERS' COMMENTS:

Reviewer #1 (Remarks to the Author):

The authors have thoroughly addressed all my comments.

Reviewer #2 (Remarks to the Author):

Thank you for your answers, explanation, and correcting the manuscript. I have no further comments.

Reviewer #3 (Remarks to the Author):

The authors have made a considerable effort in taking into account my concerns, and the other reviewers' ones, and they have clarified several previously unclear points. I think that the manuscript has been overall strengthened in this new version. I have just two further recommendations

1) Please make more explicit, in the manuscript, the limits of the current multivariate analysis of EEG data, in the very same sense as the authors do in the responses to reviewers: due to the need of averaging across really many trials, this analysis does not allow to test the Bayesian integration hypothesis in detail (for instance across all the different cue-stimulus conditions), but only at a very general level. This is already an original and remarkable outcome of the present study, but it is important to avoid an over interpretation of these nice results.

2) Some weird expressions are still present in the manuscript and a last proof-reading would be useful. To make just one example, the last sentence of the abstract reads somewhat strange: "although most of the multivariate EEG activity patterns during pursuit exhibited the retinal velocity errors accumulated over time.". What about replacing the very "exhibited" with "was best correlated with"

REVIEWERS' COMMENTS:

Reviewer #1 (Remarks to the Author):

The authors have thoroughly addressed all my comments.

Thank you for improving the manuscript with excellent comments and suggestions!

Reviewer #2 (Remarks to the Author):

Thank you for your answers, explanation, and correcting the manuscript. I have no further comments.

We appreciate the reviewer's questions and suggestions, which enhanced the manuscript a lot.

Reviewer #3 (Remarks to the Author):

The authors have made a considerable effort in taking into account my concerns, and the other reviewers' ones, and they have clarified several previously unclear points. I think that the manuscript has been overall strengthened in this new version. I have just two further recommendations

Thank you for your suggestions and comments. We were able to improve the manuscript substantially.

1) Please make more explicit, in the manuscript, the limits of the current multivariate analysis of EEG data, in the very same sense as the authors do in the responses to reviewers: due to the need of averaging across really many trials, this analysis does not allow to test the Bayesian integration hypothesis in detail (for instance across all the different cue-stimulus conditions), but only at a very general level. This is already an original and remarkable outcome of the present study, but it is important to avoid an over interpretation of these nice results.

Thank you for the important suggestion. We added the limitation of our approach in the discussion so that readers did not fall into overinterpretation (lines 421 – 424).

2) Some weird expressions are still present in the manuscript and a last proof-reading would be useful. To make just one example, the last sentence of the abstract reads somewhat strange: "although most of the multivariate EEG activity patterns during pursuit exhibited the retinal velocity errors accumulated over time.". What about replacing the very "exhibited" with "was best correlated with"

We have done the proofreading of the manuscript with the professional's help.